**Subject Category:**
Biology (whole organism)

behaviour/cognition/ecology

cognitive ecology, operational sex ratio, intraspecific competition, social context, spatial memory

**Author for correspondence:**
Alexander G. Ophir
e-mail: ophir@cornell.edu

# Social context alters spatial memory performance in free-living male prairie voles

Marissa A. Rice[1,2], Gloria Sanín[2] and Alexander G. Ophir[1,2]

[1]Department of Psychology, Cornell University, Ithaca, NY 14853, USA
[2]Department of Integrative Biology, Oklahoma State University, Stillwater, OK 74078, USA

 AGO, 0000-0002-4877-9696

Spatial memory is crucial for mating success because it enables males to locate potential mates and potential competitors in space. Intraspecific competition and its varying intensity under certain conditions are potentially important for shaping spatial memory. For example, spatial memory could enable males to know where competitors are (contest competition), it could help males find mating partners (scramble competition) or both. We manipulated the intensity of intraspecific competition in two distinct contexts by altering the operational sex ratio of prairie voles (*Microtus ochrogaster*) living in outdoor enclosures by creating male- and female-biased sex ratios. After living freely under these contexts for four weeks, we compared males' performance in a laboratory spatial memory test. Males in the male-biased context demonstrated better spatial memory performance than males in the female-biased context. Notably, these data show that in spite of experiencing equally complex *spatial* contexts (i.e. natural outdoor enclosures), it was the *social* context that influenced spatial cognition, and it did so in a manner consistent with the hypothesis that spatial memory is particularly relevant for male–male interactions.

## 1. Introduction

Spatial memory is a cognitive ability that is essential for many of life's basic challenges. For example, food caching is essential to survival for non-migrating birds and mammals that do not have access to food sources over the winter [1,2]. Particularly, plasticity in spatial cognition enables accounting for changing spatial demands in the environment [3,4]. Navigating the environment beyond using simple procedural algorithms requires building spatial representations or 'mental maps' to remember the location of salient features in the environment (e.g. food sources or territories). Surprisingly, very little attention has been paid to how spatial memory relates to social contexts (see [5]), although

some recent work has begun to consider these questions [5,6]. In particular, mating decisions are inherently spatial tasks for most animals. For instance, knowing where competitors or potential mates can be found within a social landscape should be particularly salient and relevant information with respect to acquiring mates [5,7].

Evidence has indirectly demonstrated that reliance on spatial memory among males can increase breeding competitiveness. For example, male thirteen-lined ground squirrels appear to rely on spatial memory for competitive mate searching [8]. In this species, males must search actively for females during the mating season, and their searches are biased towards locations where they previously encountered females. Similarly, male white-tailed deer that searched for mates in places where they had previously encountered females were more successful in locating mates than those that explored broader patches [9]. Cases such as these support the hypothesis that spatial cognition is an important factor for finding mating opportunities. Furthermore, spatial performance in male rodents is correlated to male–male competition. Dominant male meadow voles have better spatial learning capacity and learn faster than less dominant males [10], suggesting that spatial memory is associated with a more competitive phenotype. Thus, spatial ability is an important factor that influences mating behaviour in various spatial and social contexts, particularly ones that affect within-sex competition among males [5].

From a male perspective, there are two main components of intraspecific competition: contest competition and scramble competition [11]. Scramble competition involves males searching for females, particularly when females are widely distributed. In contest competition, males interfere with (or challenge) others' ability to access resources. One of the main resources that males are competing over is access to females or resources that would allow males to effectively attract females. To successfully secure mates, males must track the movements of females (i.e. potential mates, via scramble competition) and competitors (i.e. other males, via contest competition) in space and time through the environment. Therefore, the intensity of intraspecific competition can influence the likelihood of mate acquisition and how much males are directly competing with one another.

It is well recognized that the operational sex ratio (OSR) alters the intensity of intraspecific competition [12]. OSR can be male biased, female biased or balanced, and these biases can change over time within a population. Variation in environmental contexts such as OSR will impact the payoff matrix of the costs and benefits associated with intrasexual competition. Under male-biased OSRs, for example, contest competition should be intense because the number of male competitors is relatively high. Likewise, scramble competition (specifically males searching for potential mates) should be more intense in a female-biased OSR because the number of females is high relative to the number of males. Taken together, variation in OSR can establish social contexts in which variation in behavioural tactics (such as whether to prioritize tracking competitors versus potential mates) reflects solutions to those immediate contextual challenges. Moreover, differences in OSR have the potential to reveal the behavioural and cognitive mechanisms by which males react to different social contexts.

Here, we asked whether manipulating OSR impacts spatial cognition. More specifically, we asked how social contexts that vary the proportion of males to females created unique social challenges to which spatial cognition should respond. This is a particularly relevant challenge for prairie voles (*Microtus ochrogaster*) because natural population densities are known to fluctuate throughout the year [13]. Additionally, as males and females move through the environment, they often encounter each other (to some degree) and the majority of prairie voles will form pairs. In these cases, males and females will share and guard a nest and territory within their home range (see [14]). We hypothesized that differences in spatial memory would result from experiencing different OSRs, potentially revealing the intraspecific competitive pressures that impact spatial memory. We predicted that if male spatial memory is important for contest competition, then males experiencing a male-biased OSR should demonstrate enhanced spatial memory. Alternatively, if male spatial memory is important for acquiring mates via scramble competition, experiencing a female-biased OSR should demonstrate enhanced spatial memory. We acknowledge, *a priori*, that these are not necessarily mutually exclusive predictions. By directly comparing the spatial memory performance of males experiencing different OSRs, we sought to reveal the importance of spatial learning and memory in mating behaviours. We tested our predictions by exposing socially monogamous male prairie voles living in semi-natural field enclosures to a male-biased or female-biased OSR and then assessing spatial learning and memory in the laboratory.

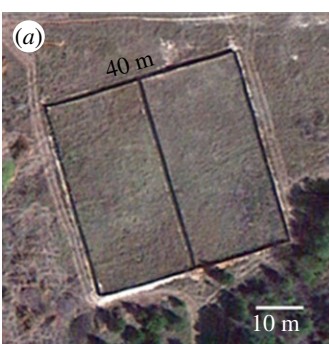 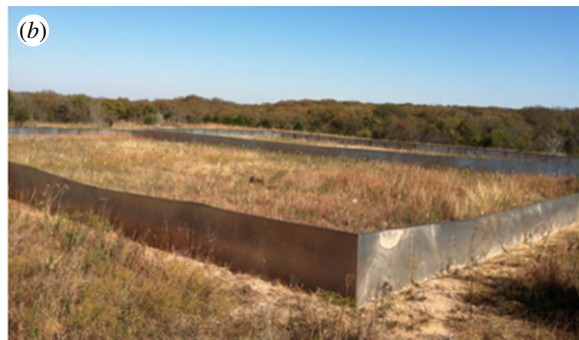

**Figure 1.** (*a*) Satellite image and (*b*) side view image of the semi-natural outdoor enclosures used in this experiment. Each adjacent enclosure measured 40 × 20 m, and was constructed of aluminium sheet metal walls and powder-coated steel tube frames. The walls extended 60 cm above, and below ground, preventing subjects from escaping (above or below) the enclosures, and preventing any other animals from getting inside.

# 2. Material and methods

## 2.1. Subjects

Animals serving in this study were the laboratory-bred offspring of 15 unrelated pairs of F1 or wild-caught prairie voles in our colony. The wild-caught breeders in our colony were originally trapped in Urbana-Champaign, IL, USA. At 21 days old, pups were weaned from parents and housed with same-sex siblings. Individuals were housed in polycarbonate cages (length 28 cm, width 18 cm, height 13 cm) and kept on a 14 : 10 h light–dark cycle. All animals were given Rodent Chow (Harland Teklad, Madison, WI, USA) and water ad libitum. We used 26 males and 24 females as subjects. All animals were sexually mature (60–90 days old) virgins. Each subject was ear-tagged with a small metal self-piercing and self-locking ear tag, laser etched with a unique numerical four digit ID. All experimental procedures were approved by the Institutional Animal Care and Use Committee at Oklahoma State University, where the behavioural work was conducted.

## 2.2. Semi-natural fieldwork

The study was performed in two adjacent enclosures located in Stillwater, Oklahoma, which is within the natural distribution of prairie vole habitat. Each enclosure measured 40 × 20 m, and was constructed of aluminium walls and powder-coated steel tube frames (figure 1). The walls extended 60 cm above and below ground, preventing subjects from escaping (above or below) the enclosures, and preventing any other animals from entering the enclosures. Both enclosures contained the same soil and vegetation, the terrain of each enclosure was nearly identical, and the distribution and composition of plants within each enclosure was similar. The vegetation consisted of dicots and mixed pasture grasses such as fescue, brome and rye, all of which are suitable natural prairie vole habitat.

Each animal (male and female) was semi-randomly assigned to either a male biased (MB) or female biased (FB) sex ratio condition. All males and females within each condition were unrelated with identical pre-existing social, genetic and developmental histories to reduce the opportunity for sampling biases. We note that we had the added control for genetic and developmental variation among females by selecting females from 12 sister-pairs and placing one sister in each treatment group. In the MB treatment, the sex ratio was 12 females and 18 males; the FB treatment was made of 12 females and 8 males. Each treatment density (males and females per unit space) was safely within the range of OSR variation naturally observed in prairie voles [13]. Importantly, the number of females was kept constant to ensure that the total number of females per unit space would be the same in each condition. We recognize that this creates a confound of overall density between enclosures; however, we were interested in the behavioural results as a consequence of males *relative* to females. For this reason, we decided to hold the total number of females in each enclosure constant. We carefully made this choice based on both empirical work in voles and theoretical work on mating systems. Classic mating system theory outlined by Emlen & Oring [7] argued that the potential for polygyny is directly impacted by the distribution of females in space. This has been strongly supported by other theoretical work (e.g. [15]). This theoretical work has provided the

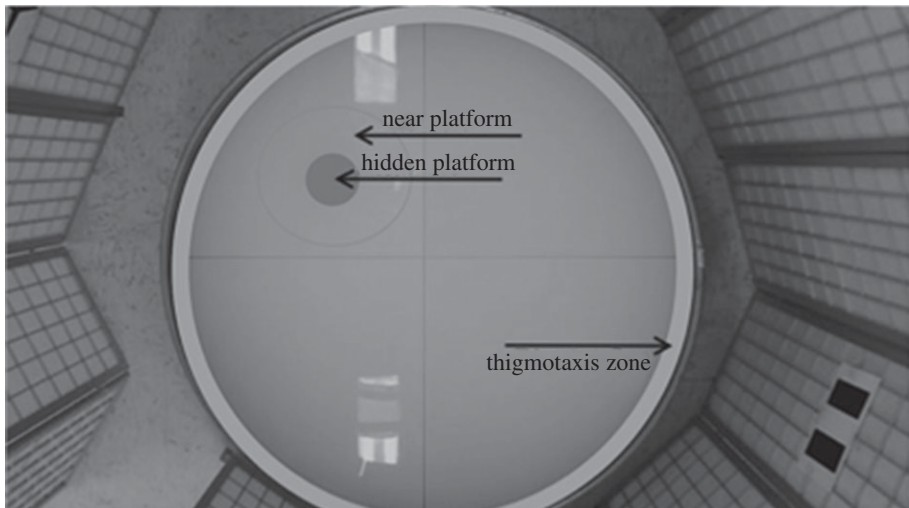

**Figure 2.** The apparatus of the Morris water maze test consisted of a 1000 l tank with a submerged platform. Quadrants and zones within the Morris water maze were used to analyse swimming performance. The zones we used were a 'thigmotaxis zone' (7.6 cm from wall, marked in light grey), which encompassed the outer edge, a 'near platform zone' (the area near the platform, 40.6 cm diameter) and a 'platform zone' (the area directly encompassing the platform itself, 11.5 cm diameter, marked in dark grey). Boundaries are superimposed over the image to outline the zones.

foundation for the idea that 'females track the environment, and males track females'. This important premise was paired with the results from Myllymäki [16], which demonstrated that female field vole distribution and home range size are impacted by food distribution (how uniform or stochastically distributed it is), and not the population density. As a result of our design, males' access to mates was determined by intraspecific competition rather than merely being of function of density.

Females were released 1 day before males were introduced into their respective enclosures. The animals were allowed to live freely for four weeks after male introduction. Next, we trapped animals from the enclosures and returned them to the laboratory. We used Fitch traps, baited with sunflower seeds and oats to trap voles. Each morning, we checked the traps along our field grid. When a vole was located inside a trap, we transferred it to a small cage and provided fresh cut apples for food and hydration. Once returned to the laboratory, the animal was given the standard resources used to maintain our laboratory vole colony (water bottle, rodent chow, nesting material, etc.). All animals were trapped within 5 days. Trapping continued for an additional 5 days (with no captures) to ensure all surviving animals had been recovered. We assumed that any untrapped animals did not survive. Of the 50 animals (26 male; 24 female) serving in this study, all but six (4 male; 2 female) were recovered. Two males from the MB and FB enclosures, and one female from each enclosure were not recovered, bringing the final OSR to 16 : 11 and 6 : 11 (M : F), respectively.

## 2.3. Behavioural testing

Immediately upon arrival to the laboratory (within 2–3 h from capture), male subjects were tested individually in the Morris water maze (figure 2), a classic test for spatial memory in which animals are repeatedly exposed to spatial cues to find a hidden target location (a platform) under water [17]. Specific details regarding the apparatus and procedure are described thoroughly in Rice *et al.* [6]. Briefly, the Morris water maze exposes animals to a large (140 cm in diameter, 1000 l) tub of opaque water, surrounded by external visual cues. An 11.5 cm platform was placed 3 cm below the surface of the water. Subjects were trained to locate the hidden platform using the spatial cues over a series of 10 2 min learning trials. Subjects were given the 10 learning trials over 5 days, 2 trials per day with an inter-trial interval of 1 h on trials conducted on the same day. In each trial, the subjects freely swam in the apparatus, and were retrieved upon successfully locating the platform. If the subjects did not locate the platform within 2 min, the subject was guided to the platform and then removed. We used the standard 2 min cut-off time for unsuccessful trials [18].

Quadrants and zones within the Morris water maze were used to analyse swimming performance. The zones we used included a 'thigmotaxis zone' (7.6 cm from wall) that encompassed the outer edge

[18], a 'near platform zone' (the area near the platform, 40.6 cm diameter) and a 'platform zone' (the area directly encompassing the platform itself, 11.5 cm diameter). We assessed spatial learning by measuring the latency to locate the platform in trials 1 to 10. We assessed spatial memory with an 11th trial, 1 h after the 10th learning trial, by removing the platform and measuring the time spent swimming in the quadrant and the area near the original location of the platform, and the frequency of visits to those locations. The spatial memory test (trial 11) lasted 1 min. We also quantified how accurately subjects were able to approximate the platform's previous location by measuring the average distance from the platform location during the memory trial.

All trials were recorded using a video camera (SR-120, Sony, New York City, NY, USA), and we used EthoVision XT 8.5 (Noldus, Leesburg, VA, USA) to automate behavioural analysis of performance in the water maze and to analyse behaviour. EthoVision is a behavioural software program that automates data collection, and we used the water maze package specific for Morris water maze tests. Unfortunately, we do not have baseline spatial cognition measures from before the treatment (i.e. the outdoor enclosure experience); however, changes in spatial memory resulting from the spatial complexity of the semi-natural enclosures relative to the standard laboratory housing all animals experienced prior to the experiment should greatly outweigh any minor initial individual variation in spatial cognition among these laboratory-bred animals.

## 2.4. Statistical analysis

A linear mixed model (LMM) with Gaussian distributed errors was used to analyse spatial learning performance in the Morris water maze by comparing the latency to reach the hidden platform for the MB and FB males. All the assumptions for the model were met. In the model, latency to platform was the response variable, while the fixed effects were 'sex bias', 'trial' and 'trial × sex bias'. Each individual subject was treated as a random effect to control for multiple responses, and the model also controlled for swim velocity and thigmotaxis by including them as fixed effects (figure 2). The F-statistics reported are from Type III sums of squares tests.

To assess the performance of MB and FB males in the spatial memory test, we compared the number of visits to, and duration of time spent swimming in, the vicinity to the platform's original location (figure 2). Duration results were compared using Student's t-test, assuming equal variance, and the number of visits or frequency results were calculated using a generalized linear model (GLM) with a Poisson distribution and a log link. Our GLM compared deviances, and was effect coded. We performed an overdispersion test and found an overdispersion parameter of 1.211. Additionally, to quantify the accuracy of each treatment group during the memory trial, we measured the average distance from the exact location of the platform and compared them using Student's t-test. Finally, we calculated Cohen's d effect sizes for all variables, which are reported in the results. All models were chosen a priori based on the experimental parameters and no model simplification was used. The LLM was chosen to account for the repeated measures of the subjects, whereas the GLM was chosen because the frequency variables are whole numbers that skewed to the right and followed the Poisson distribution. All statistics were done in JMP Pro 12 (SAS, Cary, NC, USA).

## 3. Results

In running our model to analyse spatial learning performance, we found that neither swim speed ($t_{(197)} = 0.48$, $p = 0.63$) nor thigmotaxis ($t_{(21)} = 1.22$, $p = 0.28$) had a significant effect on latency to reach the platform. However, we found a significant interaction effect of 'trial × sex bias' ($F_{(9,178)} = 1.98$, $p = 0.04$, figure 3). Specifically, MB males exhibited a significant trial effect ($F_{(9,178)} = 6.61$, $p < 0.0001$) demonstrating a marked decrease in latency to the platform throughout learning (trials 1 through 10; figure 3). Conversely, FB males showed no significant difference in latencies across the learning trials ($F_{(9,178)} = 1.65$, $p = 0.104$). Although we cannot rule out the possibility that males in the FB context demonstrated learning impairment, we note that by the end of the learning trials (trial 10), FB males found the platform 17.5% faster than the initial trial (trial 1); MB males found it 65.3% faster. Thus, males in both contexts demonstrated a consistent ability to locate the platform with improvement. However, notably, MB males clearly showed significant improvement in spatial performance over the course of learning trials, whereas FB males did not demonstrate learning to the same degree.

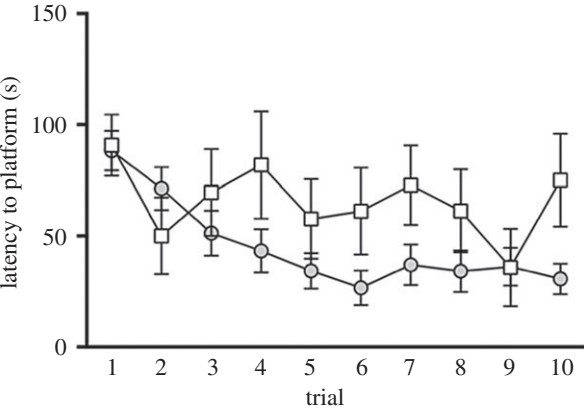

**Figure 3.** Marginal mean (±s.e.) latency in seconds (s) to reach the platform throughout all 10 learning trials for MB (grey circles, $n = 16$) males and FB (white squares, $n = 6$) males in the Morris water maze.

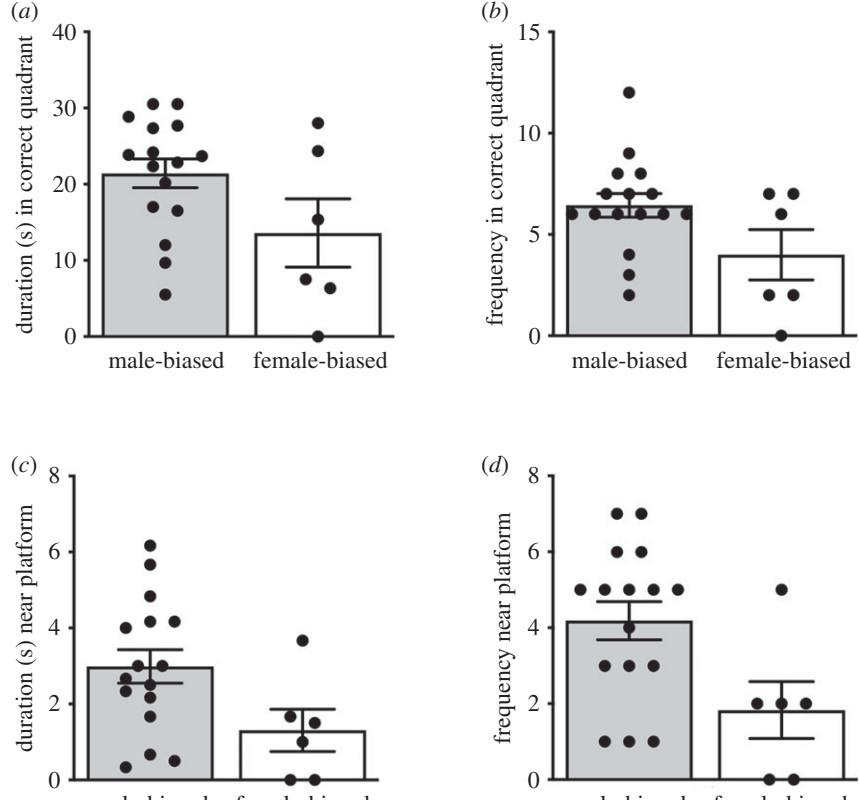

**Figure 4.** Memory trial performance. (*a*) Mean (±s.e.) time in seconds (s) subjects spent swimming in the platform-containing quadrant of the water maze, and (*b*) number of times subjects swam in the platform-containing quadrant of the water maze. (*c*) Mean (±s.e.) time in seconds (s) subjects spent swimming in the 'near platform zone', and (*d*) number of times subjects swam in the 'near platform zone' of the water maze. For (*b,d*), mean frequencies are presented; however, data were analysed using generalized linear models with a Poisson distribution and a log link (MB $n = 16$, FB $n = 6$). Dots represent individual data.

In our assessment of performance in the spatial memory test, we found that MB males tended to spend more time in the correct quadrant of the maze ($t_{(20)} = 1.93$, $p = 0.06$, $d = 0.92$ figure 4*a*) and had significantly more visits to this zone (GLM; $B = -0.237$, s.e. $= 0.113$, $p = 0.027$, $d = 0.28$, figure 4*b*) compared to males from the FB context. When refining our metric of memory performance from the gross-scale measurement of swimming in each 'quadrant' to focusing on the area proximal to the specific location of the platform, we found that MB males spent significantly more time in the area directly near the removed platform's original location ($t_{(20)} = 2.11$, $p = 0.04$, $d = 0.06$, figure 4*c*) and visited it more frequently (GLM; $B = -0.412$, s.e. $= 0.162$, $p = 0.005$, $d = 0.17$, figure 4*d*). Lastly, when we refined our measurement to assess accuracy in the memory trial to compare the average distance from

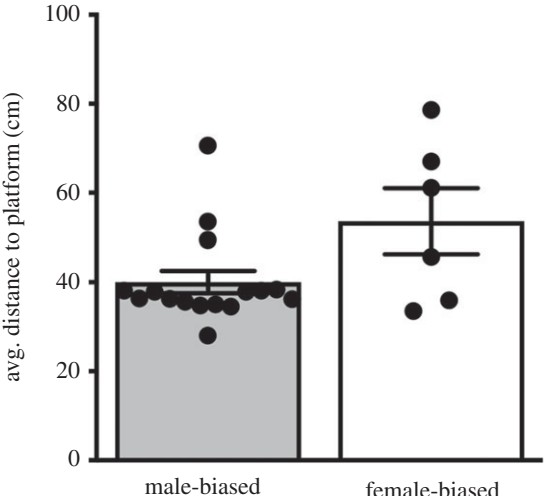

**Figure 5.** Mean (±s.e.) distance from of the swimming vole to the original location of the platform during the memory test (Trial 11) in the Morris water maze. Dots represent individual data.

the swimming vole to the specific area directly where the platform was originally located, MB males swam in closer proximity to the platform location than FB males (FB $\bar{x} = 53.64$ cm, s.e. $= 7.38$, MB $\bar{x} = 40.06$ cm, s.e. $= 2.51$; $t_{(20)} = -2.25$, $p = 0.03$, $d = 1.59$, figure 5). Taken together, our results demonstrate that the males from the MB treatment have superior spatial learning and memory, and specifically better accuracy and precision, compared to the FB treatment group.

## 4. Discussion

We found that laboratory-reared males that lived in a male-biased semi-natural social context for approximately four weeks (where males outnumbered females), out-performed males that experienced a female-biased social context (where females outnumbered males) on spatial learning and memory tasks conducted in the laboratory. The specific behavioural and cognitive demands that ecological contexts create for a population can have profound impacts on behavioural and cognitive phenotypes within that population [19,20]. Furthermore, the intensity of contest competition or scramble competition can be an important factor in shaping spatial memory. For instance, patrolling and excluding males from territories is expected to be more common as a result of increased contest competition produced by a skewed MB OSR. Alternatively, males must rely on their spatial memory to locate potential mates, particularly when females outnumber males and conditions are ripe for intense scramble competition [21]. Although these are non-mutually exclusive contexts, the results from our study suggest that only one of these contexts staged in outdoor field enclosures led to relatively improved male prairie vole spatial learning and memory based on testing in a classic laboratory assessment of spatial cognition.

Indeed, our data support the hypothesis that male prairie vole spatial memory is more heavily influenced by, and/or more sensitive to, social contexts that are likely to heighten contest competition. This result is consistent with other work that found male prairie voles attend to the social identity of males more closely than females [22]. In nature, socially monogamous males establish and presumably defend territories, while other non-monogamous males do not [14]. Ophir *et al.* [23] demonstrated that territorial 'resident' males sire more offspring than non-territorial 'wanderers'. This suggests that males' reproductive success is linked to territoriality and possibly mate guarding. If males compete for territories with each other, and territoriality leads to increased reproductive success, then male spatial distribution should be highly valuable information to male prairie voles. The adaptive specialization hypothesis states that cognition is specialized to solve specific ecological problems [19]. It is well recognized that spatial ability is plastic and responsive to environmental complexity [24–26]. Possible mechanisms for the changes we observed following living in the field for such a short time period (four weeks) include structural plasticity, such as hippocampal dendritic growth, cell proliferation and/or adult neurogenesis [4]. In this case, flexibility and plasticity of spatial ability appears to be important for male prairie vole mating behaviours in certain contexts more than

others. Therefore, spatial flexibility in performance might enable animals to respond to their social context in a manner that is most advantageous within their immediate situation [19]. The form that such a spatially flexible response should take could vary depending on the behavioural ecology and natural history of the species in question. In prairie voles specifically, proximate pressures such as social landscapes in which males outnumber females (i.e. a male-biased OSR) could impact male spatial ability to enable defending own territories and possibly tracking the territories of other competitors. It has been proposed that socially monogamous male prairie voles maximize mate guarding to protect against cuckoldry by tracking other males' location across the social landscape and actively excluding them from gaining access to female partners [5,27,28]. The results reported here are the first data that support this notion. Taken together, these studies indicate that male mating tactics, and the associated cognitive tools that support them, are heavily influenced by male–male (contest) competition.

## 4.1. An effect of sex ratio or population density?

Our experiment was designed to ask if males' (memory-based) behaviour would be altered by the social landscape. To this end, we were faced with two design alternatives: (i) to either alter the total population density but keep the total number of females constant or (ii) to keep the total population density constant and alter the number of males and females relative to each other. We chose the former rather than the latter for several important reasons. Firstly, classic literature on mating system theory outlined by Emlen & Oring [7], and supported by Shuster & Wade [15], argues that male mating tactics are directly impacted by the distribution of females in space. The idea that 'females track the environment, and males track females' has also received experimental support. For example, Myllymäki [16] demonstrated that female field vole distribution and home range size are impacted by food distribution in the environment, and not the population density. Therefore, the available work (both empirical and theoretical) strongly supports the interpretation that males react to availability of females relative to males, independently of the total population density. Secondly, and perhaps more importantly, keeping the total population density the same would necessarily alter the number of females per enclosure and context. Altering the absolute number of females would have had the unintended consequence of changing the amount of space each female was able to occupy. Such a manipulation would have introduced a problematic confound between contexts; more females in a fixed-sized enclosure would provide less space per female compared to the alternative context. We acknowledge the possibility that male behaviour could have been impacted by population density in addition or instead of the sex ratio. However, with the aforementioned supporting knowledge that females' distributions should not be impacted by population density, we felt confident that any changes in male spatial ability would be more likely to be attributable to their responses to the sex ratio, which would be best assessed by fixing the number of females per unit space. Therefore, we deliberately held constant the number of females so that we could evaluate how the impact of sex ratio would alter behaviour in contexts in which the same number of females was distributed over the same amount of space. Ultimately, we prioritized accepting the potential (but unlikely) confound of population density over the potential confound of altered females per unit space (i.e. holding an independent variable constant) because our experiment focused on space use issues as our dependent variable. Nevertheless, it will be important for future work to alter population density (with same and altered sex ratios) to fully substantiate our interpretation that the sex ratio is the key factor in explaining how social context altered male spatial ability.

## 4.2. Social context can drive cognitive plasticity

Our study demonstrates an important and well-recognized point about the nature of spatial learning and memory: it is highly flexible and sensitive to experience and the environment. Traditionally, plasticity in spatial memory has been demonstrated in experiments that manipulated physical environmental enrichment. For example, living in an enriched environment improves spatial learning in the Morris water maze [29] and spatial working memory [30] in rats. In our study, we placed animals in highly complex and enriched environments (semi-natural outdoor field enclosures), which could certainly be expected to have modified the spatial memory abilities of our subjects. Indeed, all of the animals in our experiment were equally exposed to the challenges of living in nature, including experiencing temperature fluctuations and a need to locate shelter, the need to explore their environments and forage for food, and experiencing diverse terrain with a heterogeneous mix of vegetation and spatial

landmarks. These challenges experienced in the field were much more complex, and in particular, profoundly more spatially complex than standard laboratory housing. However, if physical features of environmental complexity alone accounted for altered spatial memory, the performance of our subjects should have been uniformly strong across social contexts. Yet, males in the MB context demonstrated superior spatial ability compared to FB males. Social complexity is an important form of environmental complexity, even if it is not commonly discussed as such. Remarkably, our results are consistent with the hypothesis that spatial learning and memory was impacted by the social context, rather than the spatial context.

# 5. Conclusion

We took an integrative approach to understand the cognitive ecology of spatial memory as an outcome of living in real-world socially and spatially complex environments. In sum, we altered sex ratio to determine if spatial cognition would respond to proximal social pressures, and found that males under contest competition were more adept at spatial learning and memory compared to males living under a scramble competition context. Our data indicate that social complexity, potentially driven by contest competition, alters spatial memory performance. Indeed, the salient feature that differed between subjects, and accounted for different performance in the spatial memory task was the proportion of males to females within each enclosure that males experienced. That is, spatial cognition was responsive to alterations of the OSR context within highly spatially complex environments. To our knowledge, this is the first demonstration in which spatial complexity was high and held constant under natural field conditions, and the composition of the social context impacted cognitive functions related to spatial information.

Ethics. All procedures were approved by the Institutional Animal Care and Use Committee (IACUC: AS096) of the sponsoring institution and consistent with guidelines for the ethical use of animals in research.

Data accessibility. The raw data associated with this manuscript are available at the Dryad Digital Repository: https://dx.doi.org/10.5061/dryad.0vt4b8gtp [31].

Authors' contributions. M.A.R. conceptualized the experiment, performed the behavioural testing, analysed data and wrote the manuscript. G.S. assisted in behavioural trials and behavioural analyses. A.G.O. conceptualized the experiment, developed the experimental design, analysed data and wrote the manuscript.

Competing interests. A.G.O. is an Associate Editor of Royal Society Open Science.

Funding. This work was supported by funding from the National Science Foundation (grant no. 1354760 to A.G.O. and grant no. 2012142934 to M.A.R.), Oklahoma State University Biological Basis of Human and Animal Behavior REU Program (grant no. 1063091) that supported G.S. and the National Institute of Child Health and Human Development (grant nos. HD065604 and HD079573 to A.G.O.).

Acknowledgements. We thank Adam Dyess for his assistance in maintaining the field enclosures in Oklahoma. We also thank Stephen Parry at the Cornell Statistical Consulting Unit for statistical input regarding our analyses. Finally, we thank the four anonymous reviewers for their thoughtful and constructive comments, which improved this manuscript.

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
