## [Reviewer comments · Royal Society Open Science]

Review History

RSOS-190743.R0 (Original submission)

Review form: Reviewer 1

Is the manuscript scientifically sound in its present form?

Yes

Are the interpretations and conclusions justified by the results?

No

Is the language acceptable?

Yes

Is it clear how to access all supporting data?

Yes

Do you have any ethical concerns with this paper?

No

Have you any concerns about statistical analyses in this paper?

No

Recommendation?

Major revision is needed (please make suggestions in comments)

Comments to the Author(s)

This is a relatively short and well written paper comparing the spatial memory of male voles that had experience in two different experimental sex-ratio conditions. This study builds off of the vast amount of work suggesting that the physical environment has an effect on memory, both at the ontogenetic and phylogenetic levels. One of the overarching goals of the paper was to demonstrate that the social environment also plays a role in the creation/manipulation of memory. The manuscript is clear and is for the most part fairly straightforward. The topic is current, very interesting, and has the potential to make an important impact on the field. We certainly do need to consider the role of both the social and physical environments when studying the formation and maintenance of memory.

1. Although I really like the paper, the main weakness, in my opinion, is in its interpretation of the results given the nature of the experimental design. The authors manipulated the sex ratio of the experimental groups by keeping the number of females constant and changing the number of males, thereby changing the competition among males for access to females. In doing so, they also manipulated vole density and thereby changed the competition among all of the voles for resources. Although the authors acknowledge this confound, they spend virtually no additional time discussing it or addressing its possible effect on their conclusions. The problem here is that the directions of the effect on competition are equal for both the sex competition and general resource competition hypotheses. Therefore, the authors cannot conclude that the change in memory in the males was necessarily a function of the change in competition for females.

I do not think that this confound is a deal-breaker, per se. The authors have important evidence that memory capabilities can be affected relatively quickly by changes in social conditions (and that the memory for one situation can apply to other situations, so based on ability to learn/form memory rather than simply the amount that has been learned). However, the authors need to spend much more time considering alternative hypotheses, especially in the discussion. Given the experiment, I see a place for both ideas (sex ratio and general resource competition) being interpreted as the effect of the social environment on memory. In addition, some discussion about the possible mechanisms for this effect (e.g., change in CORT, neurogenesis) could help contextualize the effect, which might be helpful for generating future hypotheses and predictions.

If the authors have any evidence of the memory abilities of the females from this experiment or other mechanistic data to support the changes in memory, those would be nice to include (or at least reference) here.

2. Methods. The authors should include some additional information about the transfer of the voles from Illinois to Oklahoma. This should be done for two reasons. First, it was not particularly clear that they were from two different areas. I missed this the first few times. So, I suggest making this point specific. Second, a note about the similarity and/or differences in habitat between the sites is important. This is a minor point, but something that would be appreciated for clarity.

3. Lines 118. This point relates in part to point 1 above. The argument here suggests that female home range size is not affected by female density. I find this hard to accept from an ecological perspective. Perhaps at low to moderate densities, home range size would not be affected;

however, at very high densities there would be some effect, if for no other reason than access to resources. This should be clarified or nuanced. Moreover, within the context of the experiment and densities therein, this point may stand for females, but the females are not the subjects of the study. What about males? How are male home ranges expected to change with overall changing vole densities?

4. Figure 3. It seems that the males in the Female-Biased group did not learn the water maze. The time to solve the maze did not decrease over the course of the trials. This is a problem. It is odd that they could not or did not learn at all. Is there some other explanation for why this inability to learn was occurring? I buy the idea that those males in the Male-Biased group would need to use and hone their memory and so might do better on subsequent memory tests. However, I find it suspicious that the Female-Biased group showed no learning at all. In addition, why did the authors stop at 10 trials? Why test the Female-Biased group when they were not at criterion?

5. Figure 4. I am having difficulty resolving the lack of learning in the Female-Biased group (from Figure 3) with the very small differences in time spent in the quadrant/platform areas indicative of spatial memory/learning of this figure. The mean difference between the groups is only about 5 seconds of time in (Fig 4a) and 2 visits to (Fig 4b) the correct quadrant and 1.5 seconds near (Fig 4c) and 2 visits to (Fig 4c) the platform area. These differences seem tiny relative to the context of 120 seconds (max time) in the maze and the average of 50 seconds difference in time to find the platform (Figure 3).

Review form: Reviewer 2

Is the manuscript scientifically sound in its present form?

Yes

Are the interpretations and conclusions justified by the results?

No

Is the language acceptable?

Yes

Is it clear how to access all supporting data?

No

Do you have any ethical concerns with this paper?

No

Have you any concerns about statistical analyses in this paper?

No

Recommendation?

Major revision is needed (please make suggestions in comments)

Comments to the Author(s)

Adult prairie voles were released into semi-natural enclosures with either a male-biased (MB) or female-biased (FB) sex ratio (SR) for a period of four weeks, after which they were re-captured and males were tested in the Morris water maze to examine spatial learning and memory. After the four-week treatment MB males showed significant improvement in latency to locate the

hidden platform over successive trials in the water maze while FB males did not; MB males also visited more often and spent more time in the correct escape zone during the probe trial. This experiment is simple and has provided clear, consistent results, but has not properly tested what the authors claim to have tested. While sex ratio is the treatment effect of interest, it is not sex-ratio per se, but rather the effects of male density that are tested in this experiment. As such, I think the background, objectives, hypotheses, and discussion could use substantial revision.

Missing from the introduction are the reasons why SR (or rather, male density) might influence male cognitive abilities after four weeks. While it is clear that natural selection can shape spatial learning and memory, it is not clear why the environment an adult experience would influence their spatial abilities. The study on male meadow voles suggests intraspecific variation in a congener but does not indicate why this phenotypic link exists. Some details are provided in the discussion (L214-229) that could be incorporated into the introduction. As is, the theory supporting the hypothesis that SR influences an individual's spatial cognition are not clearly laid out. What mechanism could be at work over a four-week period?

As mentioned, the hypothesis tested is not quite what is stated: while both scramble and interference competition were mentioned, the design of this experiment only allows for examination of the effects of interference competition on male cognitive abilities. Because there are 12 females in each treatment there is no way to compare spatial abilities relative to differences in scramble competition. Rather, this experiment is only testing the effects of male density, and if differences in male density and the hypothesized competition inherent in lower vs. higher male density affect male's spatial learning and memory abilities.

In regards to the results and discussion, it is again important to note that the effects of male-biased sex ratio cannot be separated from the effects of density. To separate the effects of sex ratio and density, enclosures with even sex ratios but density equivalent to the enclosures used in this experiment would have provided crucial, proper controls. This would allow for the effects of SR itself to be examined.

Specific comments:

L15- 'time' how is this related to spatial memory? This would be time/place learning. You can have spatial learning without a time component. Please be specific. Did you test time learning?

L23-4- Why are there difference, then, between the MB and FB conditions? What is driving the differences? I would remove this statement because we do not know what 'complex' means for these animals.

L29- change cognitive "feature" to "ability"

L32- navigation does not necessarily require spatial representations or "mental maps", e.g. step-counting

L46- 'episodic-like memory' What is this and why is it important to the current experiment?

L50- use 'performance' instead of 'capacity'

L52- confusing as I thought the authors have been trying to articulate that mating behaviour is an important factor in determining spatial memory?!?

L111-Could they hear vocalizations from the other enclosure - what potential effect could this have?

L118- 1977 reference - has this statement regarding distribution/home range size been verified in prairie voles? Density does seem to at least affect social associations (see Streatfeild et al. 2011 *Animal Behaviour*).

L124- Please describe how animals were trapped. The trapping period to being animals back to the lab was up to 25% of the duration of the treatments. How could this affect your results? Was date of trap (and resultant SR [and density]) accounted for in the analyses?

L183-5 This does not belong in the results

L212- the adaptive specialization hypothesis does not apply to individual cognitive plasticity does it?

L215- also see Buchanan et al. 2013 *Trends in Ecology and Evolution*

L251- Conclusion paragraph highly repetitive with previous paragraph.

Discussion- Did you measure behaviour? Do you have any idea what the males in the different treatments were doing?

Fig. 3 & 4- perhaps add sample sizes to figure, either in figure captions or in figures themselves

Review form: Reviewer 3

Is the manuscript scientifically sound in its present form?

Yes

Are the interpretations and conclusions justified by the results?

No

Is the language acceptable?

Yes

Do you have any ethical concerns with this paper?

No

Have you any concerns about statistical analyses in this paper?

Yes

Recommendation?

Major revision is needed (please make suggestions in comments)

Comments to the Author(s)

Vole spatial memory review

This is a very interesting study that examines how social context experienced by male prairie voles influences their spatial memory. Animals experience fluctuations in operational sex ratio (the ratio of males to females in a population) and this can influence males in the population by affecting the number of competitors that are trying to find females or the number of other males that may interfere with their mating behaviour. The authors put male prairie voles into one enclosure with a female-biased operational sex ratio (40% males) and other voles in a separate enclosure with a male-biased operational sex ratio (60% males). At the end of the 4 week period, they tested the spatial memory performance of males using a Morris water maze. The authors show that male voles from the enclosure with the male-biased operational sex ratio had higher spatial memory than those from the enclosure with a female-biased operational sex ratio.

These results are very interesting as the authors note that such studies investigating the role of the social environment and how it might influence changes in spatial memory are quite rare (especially in animals living in natural or semi-natural environments). The manuscript is clear and well-presented and I have only a few questions about the statistical analyses (listed below). I do have some questions and request for elaboration on a few points below. Overall, I think this is a very interesting study but the authors just need to address a few additional points and discuss some of the limitations.

Major comments:

The authors present results from one enclosure where voles were housed at a female-biased OSR and one enclosure where the voles were housed at a male-biased enclosure. Thus, there are some limitations that might be mentioned in the Discussion such as 1) they effectively have one

statistical replicate of each treatment but are using individuals within each of the enclosures as their unit of replication and 2) they do not have data on male spatial memory performance in enclosures with an equal OSR. I think it would be helpful here to make note of the unit of replication here and its limitations and also make clear to readers why they think the different OSR in the one enclosure is the actual factor causing the difference and not some other feature that differs between the two enclosures. Second, I think it would be helpful if the authors indicated what they would expect regarding male spatial memory performance in enclosures with equal OSR (perhaps in the Introduction and acknowledge that this wasn't performed and why). This comes up again in the Discussion (e.g., lines 203-205) where the authors interpret their results but it is difficult to do so without a third treatment of equal OSR.

A third limitation from the study design is that the authors had two treatments that varied the proportion of males in each enclosure but the density of voles also differed. Specifically, the male-biased OSR enclosure was 18 males and 12 females (60% males) and the female-biased OSR was 8 males and 12 females (40% males). It would be useful for the authors to note whether these treatments are within the natural range of variation in OSR – are they similar to what is found in nature? Second, this creates a confound with density where vole density in the male-biased OSR is 375 voles/ha and 250 voles/ha in the female-biased enclosure (so 50% higher in the male-biased enclosure). Similar to the above, the authors should note the densities here and whether these are within the natural range of variation. Although the authors acknowledge that this difference in density complicates the analyses in the Methods section (line 114), I think they need to provide greater discussion as it is not clear to me how they can differentiate the effect of density vs. OSR. One thought is that male home range sizes may vary as a function of density (is this known in this species?) and males in the male-biased OSR with higher density might have lower home range sizes. We therefore might expect that males in the male-biased OSR would have poorer spatial memory abilities and thus the results presented here (that males in the male-biased OSR actually have higher spatial memory) do not support that prediction.

It is fascinating that in only 4 weeks of exposure to a male-biased population sex ratio, male voles exhibited greater spatial memory performance. However, one of the notable missing pieces of data is a “before treatment” evaluation of their spatial memory performance. This should be discussed.

I'd like the authors to provide a greater context on their results. How quickly can spatial memory change in response to adjustments in the social context in other lab studies? This is not a major criticism. I just find the 4 week time period to be intriguing and would like to know more and the authors could help readers here by providing some context.

The authors might broaden the manuscript by focusing on some of the underlying machinery that drives the generation and storage of these “mental maps” that are influenced by social context. Perhaps in the Introduction it would be helpful to give readers an idea of what the current state of our knowledge is regarding how these maps are created and stored?

I think the authors should move towards a better way of visualizing the data (Figs. 3 and 4) and show all the data. Currently the data are presented in a way that only allows readers to see means and SE, which many agree mask important individual variation (e.g., Weissgerber et al., 2015 PLoS Biology). I would encourage the authors to change the way the data are presented so readers can make assessments of the data themselves and visualize individual-variation.

Other minor comments:

Line 85 – to provide readers with greater understanding of the motivation of this experiment, could you indicate how often OSR in prairie voles fluctuates in natural populations? Is it likely

that male prairie voles even experience female- and male-biased OSR in nature? If they do experience a male-biased OSR, couldn't they just leave that area to find another?

Line 109 – are there any quantitative measures to show that the vegetation in the enclosures was the same?

Line 124 – could you give a few more details on the trapping schedule used here? Randomly distributed traps?

Line 127 – here the location of those animals that were not recovered should be provided. In addition, the authors should indicate what the vole OSR and density was in each enclosure at the end of the 4 week experiment.

Line 131 – are there data on how long it took to get the males from the lab to do this test?

Line 136 – is this thigmotaxis zone based upon some previous study? It seems arbitrary but this may just be my lack of familiarity here.

Line 153 – could you indicate 1) how many observers extracted behavioral data from the videos and 2) whether they were blind to the treatments?

Line 161- I'm not quite understanding why sex is in this model. Were females tested?

Line 162 – these should all be t-statistics since the numerator df are 1?

Line 166 – it would be helpful here to say something like “at the end of the trials, males from the MB found the platform x% faster than at the beginning whereas males in the FB enclosures ...” just to put some biological reality on the results.

Line 174 – this generalized linear model needs to report the dispersion parameter and whether or not it was overdispersed.

Line 202 – but isn't it difficult to test either of these hypotheses without a third treatment where OSR was equal? Similarly, the lack of multiple replicates for the OSR treatments is a major confound.

Review form: Reviewer 4 (Ellis Langley)

Is the manuscript scientifically sound in its present form?

No

Are the interpretations and conclusions justified by the results?

No

Is the language acceptable?

Yes

Is it clear how to access all supporting data?

Yes

Do you have any ethical concerns with this paper?

No

Have you any concerns about statistical analyses in this paper?

Yes

Recommendation?

Major revision is needed (please make suggestions in comments)

Comments to the Author(s)

Please see attached file (Appendix A).

Decision letter (RSOS-190743.R0)

28-Jun-2019

Dear Dr Ophir,

The editors assigned to your paper ("Social context alters spatial memory performance in free-living male prairie voles (*Microtus ochrogaster*)") have now received comments from reviewers. We would like you to revise your paper in accordance with the referee and Associate Editor suggestions which can be found below (not including confidential reports to the Editor). Please note this decision does not guarantee eventual acceptance.

Please submit a copy of your revised paper before 21-Jul-2019. Please note that the revision deadline will expire at 00.00am on this date. If we do not hear from you within this time then it will be assumed that the paper has been withdrawn. In exceptional circumstances, extensions may be possible if agreed with the Editorial Office in advance. We do not allow multiple rounds of revision so we urge you to make every effort to fully address all of the comments at this stage. If deemed necessary by the Editors, your manuscript will be sent back to one or more of the original reviewers for assessment. If the original reviewers are not available, we may invite new reviewers.

If your study uses humans or animals please include details of the ethical approval received, including the name of the committee that granted approval. For human studies please also detail

whether informed consent was obtained. For field studies on animals please include details of all permissions, licences and/or approvals granted to carry out the fieldwork.

- Data accessibility

If you wish to submit your supporting data or code to Dryad (<http://datadryad.org/>), or modify your current submission to dryad, please use the following link:
<http://datadryad.org/submit?journalID=RSOS&manu=RSOS-190743>

- Competing interests

- Authors' contributions

- Acknowledgements

- Funding statement

Kind regards,
Lianne Parkhouse
Editorial Coordinator

on behalf of Dr Ryan Y Wong (Associate Editor) and Kevin Padian (Subject Editor)
openscience@royalsociety.org

Subject Editor's comments (Professor Kevin Padian):

Hi Alex, thanks for submitting and I hope the comments are useful in your revision. Best of luck!

Associate Editor's comments (Dr Ryan Y Wong):

Dear Dr. Ophir,

Your manuscript has been reviewed by four reviewers. Overall the reviewers found the study to be interesting and makes a useful contribution to the field. However, there were several concerns regarding limitations of the experimental design, scope of interpretations of results given experimental design, and insufficient details for statistical analyses. I recommend a Major Revision before further consideration.

Reviewers' Comments to Author:

Reviewer: 1

This is a relatively short and well written paper comparing the spatial memory of male voles that had experience in two different experimental sex-ratio conditions. This study builds off of the vast amount of work suggesting that the physical environment has an effect on memory, both at the ontogenetic and phylogenetic levels. One of the overarching goals of the paper was to demonstrate that the social environment also plays a role in the creation/manipulation of memory. The manuscript is clear and is for the most part fairly straightforward. The topic is current, very interesting, and has the potential to make an important impact on the field. We certainly do need to consider the role of both the social and physical environments when studying the formation and maintenance of memory.

1. Although I really like the paper, the main weakness, in my opinion, is in its interpretation of the results given the nature of the experimental design. The authors manipulated the sex ratio of the experimental groups by keeping the number of females constant and changing the number of males, thereby changing the competition among males for access to females. In doing so, they also manipulated vole density and thereby changed the competition among all of the voles for resources. Although the authors acknowledge this confound, they spend virtually no additional time discussing it or addressing its possible effect on their conclusions. The problem here is that the directions of the effect on competition are equal for both the sex competition and general resource competition hypotheses. Therefore, the authors cannot conclude that the change in memory in the males was necessarily a function of the change in competition for females.

I do not think that this confound is a deal-breaker, per se. The authors have important evidence that memory capabilities can be affected relatively quickly by changes in social conditions (and that the memory for one situation can apply to other situations, so based on ability to learn/form memory rather than simply the amount that has been learned). However, the authors need to

spend much more time considering alternative hypotheses, especially in the discussion. Given the experiment, I see a place for both ideas (sex ratio and general resource competition) being interpreted as the effect of the social environment on memory. In addition, some discussion about the possible mechanisms for this effect (e.g., change in CORT, neurogenesis) could help contextualize the effect, which might be helpful for generating future hypotheses and predictions.

If the authors have any evidence of the memory abilities of the females from this experiment or other mechanistic data to support the changes in memory, those would be nice to include (or at least reference) here.

2. Methods. The authors should include some additional information about the transfer of the voles from Illinois to Oklahoma. This should be done for two reasons. First, it was not particularly clear that they were from two different areas. I missed this the first few times. So, I suggest making this point specific. Second, a note about the similarity and/or differences in habitat between the sites is important. This is a minor point, but something that would be appreciated for clarity.

3. Lines 118. This point relates in part to point 1 above. The argument here suggests that female home range size is not affected by female density. I find this hard to accept from an ecological perspective. Perhaps at low to moderate densities, home range size would not be affected; however, at very high densities there would be some effect, if for no other reason than access to resources. This should be clarified or nuanced. Moreover, within the context of the experiment and densities therein, this point may stand for females, but the females are not the subjects of the study. What about males? How are male home ranges expected to change with overall changing vole densities?

4. Figure 3. It seems that the males in the Female-Biased group did not learn the water maze. The time to solve the maze did not decrease over the course of the trials. This is a problem. It is odd that they could not or did not learn at all. Is there some other explanation for why this inability to learn was occurring? I buy the idea that those males in the Male-Biased group would need to use and hone their memory and so might do better on subsequent memory tests. However, I find it suspicious that the Female-Biased group showed no learning at all. In addition, why did the authors stop at 10 trials? Why test the Female-Biased group when they were not at criterion?

5. Figure 4. I am having difficulty resolving the lack of learning in the Female-Biased group (from Figure 3) with the very small differences in time spent in the quadrant/platform areas indicative of spatial memory/learning of this figure. The mean difference between the groups is only about 5 seconds of time in (Fig 4a) and 2 visits to (Fig 4b) the correct quadrant and 1.5 seconds near (Fig 4c) and 2 visits to (Fig 4c) the platform area. These differences seem tiny relative to the context of 120 seconds (max time) in the maze and the average of 50 seconds difference in time to find the platform (Figure 3).

Reviewer: 2

Adult prairie voles were released into semi-natural enclosures with either a male-biased (MB) or female-biased (FB) sex ratio (SR) for a period of four weeks, after which they were re-captured and males were tested in the Morris water maze to examine spatial learning and memory. After the four-week treatment MB males showed significant improvement in latency to locate the hidden platform over successive trials in the water maze while FB males did not; MB males also visited more often and spent more time in the correct escape zone during the probe trial. This experiment is simple and has provided clear, consistent results, but has not properly tested what the authors claim to have tested. While sex ratio is the treatment effect of interest, it is not sex-

ratio per se, but rather the effects of male density that are tested in this experiment. As such, I think the background, objectives, hypotheses, and discussion could use substantial revision. Missing from the introduction are the reasons why SR (or rather, male density) might influence male cognitive abilities after four weeks. While it is clear that natural selection can shape spatial learning and memory, it is not clear why the environment an adult experience would influence their spatial abilities. The study on male meadow voles suggests intraspecific variation in a congener but does not indicate why this phenotypic link exists. Some details are provided in the discussion (L214-229) that could be incorporated into the introduction. As is, the theory supporting the hypothesis that SR influences an individual's spatial cognition are not clearly laid out. What mechanism could be at work over a four-week period?

As mentioned, the hypothesis tested is not quite what is stated: while both scramble and interference competition were mentioned, the design of this experiment only allows for examination of the effects of interference competition on male cognitive abilities. Because there are 12 females in each treatment there is no way to compare spatial abilities relative to differences in scramble competition. Rather, this experiment is only testing the effects of male density, and if differences in male density and the hypothesized competition inherent in lower vs. higher male density affect male's spatial learning and memory abilities.

In regards to the results and discussion, it is again important to note that the effects of male-biased sex ratio cannot be separated from the effects of density. To separate the effects of sex ratio and density, enclosures with even sex ratios but density equivalent to the enclosures used in this experiment would have provided crucial, proper controls. This would allow for the effects of SR itself to be examined.

Specific comments:

L15- 'time' how is this related to spatial memory? This would be time/place learning. You can have spatial learning without a time component. Please be specific. Did you test time learning?

L23-4- Why are there difference, then, between the MB and FB conditions? What is driving the differences? I would remove this statement because we do not know what 'complex' means for these animals.

L29- change cognitive "feature" to "ability"

L32- navigation does not necessarily require spatial representations or "mental maps", e.g. step-counting

L46- 'episodic-like memory' What is this and why is it important to the current experiment?

L50- use 'performance' instead of 'capacity'

L52- confusing as I thought the authors have been trying to articulate that mating behaviour is an important factor in determining spatial memory?!

L111-Could they hear vocalizations from the other enclosure - what potential effect could this have?

L118- 1977 reference - has this statement regarding distribution/home range size been verified in prairie voles? Density does seem to at least affect social associations (see Streatfeild et al. 2011 *Animal Behaviour*).

L124- Please describe how animals were trapped. The trapping period to being animals back to the lab was up to 25% of the duration of the treatments. How could this affect your results? Was date of trap (and resultant SR [and density]) accounted for in the analyses?

L183-5 This does not belong in the results

L212- the adaptive specialization hypothesis does not apply to individual cognitive plasticity does it?

L215- also see Buchanan et al. 2013 *Trends in Ecology and Evolution*

L251- Conclusion paragraph highly repetitive with previous paragraph.

Discussion- Did you measure behaviour? Do you have any idea what the males in the different treatments were doing?

Fig. 3 & 4- perhaps add sample sizes to figure, either in figure captions or in figures themselves

Reviewer: 3

Vole spatial memory review

This is a very interesting study that examines how social context experienced by male prairie voles influences their spatial memory. Animals experience fluctuations in operational sex ratio (the ratio of males to females in a population) and this can influence males in the population by affecting the number of competitors that are trying to find females or the number of other males that may interfere with their mating behaviour. The authors put male prairie voles into one enclosure with a female-biased operational sex ratio (40% males) and other voles in a separate enclosure with a male-biased operational sex ratio (60% males). At the end of the 4 week period, they tested the spatial memory performance of males using a Morris water maze. The authors show that male voles from the enclosure with the male-biased operational sex ratio had higher spatial memory than those from the enclosure with a female-biased operational sex ratio.

These results are very interesting as the authors note that such studies investigating the role of the social environment and how it might influence changes in spatial memory are quite rare (especially in animals living in natural or semi-natural environments). The manuscript is clear and well-presented and I have only a few questions about the statistical analyses (listed below). I do have some questions and request for elaboration on a few points below. Overall, I think this is a very interesting study but the authors just need to address a few additional points and discuss some of the limitations.

Major comments:

The authors present results from one enclosure where voles were housed at a female-biased OSR and one enclosure where the voles were housed at a male-biased enclosure. Thus, there are some limitations that might be mentioned in the Discussion such as 1) they effectively have one statistical replicate of each treatment but are using individuals within each of the enclosures as their unit of replication and 2) they do not have data on male spatial memory performance in enclosures with an equal OSR. I think it would be helpful here to make note of the unit of replication here and its limitations and also make clear to readers why they think the different OSR in the one enclosure is the actual factor causing the difference and not some other feature that differs between the two enclosures. Second, I think it would be helpful if the authors indicated what they would expect regarding male spatial memory performance in enclosures with equal OSR (perhaps in the Introduction and acknowledge that this wasn't performed and why). This comes up again in the Discussion (e.g., lines 203-205) where the authors interpret their results but it is difficult to do so without a third treatment of equal OSR.

A third limitation from the study design is that the authors had two treatments that varied the proportion of males in each enclosure but the density of voles also differed. Specifically, the male-biased OSR enclosure was 18 males and 12 females (60% males) and the female-biased OSR was 8 males and 12 females (40% males). It would be useful for the authors to note whether these treatments are within the natural range of variation in OSR – are they similar to what is found in nature? Second, this creates a confound with density where vole density in the male-biased OSR is 375 voles/ha and 250 voles/ha in the female-biased enclosure (so 50% higher in the male-biased enclosure). Similar to the above, the authors should note the densities here and whether these are within the natural range of variation. Although the authors acknowledge that this difference in density complicates the analyses in the Methods section (line 114), I think they need to provide greater discussion as it is not clear to me how they can differentiate the effect of density vs. OSR. One thought is that male home range sizes may vary as a function of density (is

this known in this species?) and males in the male-biased OSR with higher density might have lower home range sizes. We therefore might expect that males in the male-biased OSR would have poorer spatial memory abilities and thus the results presented here (that males in the male-biased OSR actually have higher spatial memory) do not support that prediction.

It is fascinating that in only 4 weeks of exposure to a male-biased population sex ratio, male voles exhibited greater spatial memory performance. However, one of the notable missing pieces of data is a “before treatment” evaluation of their spatial memory performance. This should be discussed.

I’d like the authors to provide a greater context on their results. How quickly can spatial memory change in response to adjustments in the social context in other lab studies? This is not a major criticism. I just find the 4 week time period to be intriguing and would like to know more and the authors could help readers here by providing some context.

The authors might broaden the manuscript by focusing on some of the underlying machinery that drives the generation and storage of these “mental maps” that are influenced by social context. Perhaps in the Introduction it would be helpful to give readers an idea of what the current state of our knowledge is regarding how these maps are created and stored?

I think the authors should move towards a better way of visualizing the data (Figs. 3 and 4) and show all the data. Currently the data are presented in a way that only allows readers to see means and SE, which many agree mask important individual variation (e.g., Weissgerber et al., 2015 PLoS Biology). I would encourage the authors to change the way the data are presented so readers can make assessments of the data themselves and visualize individual-variation.

Other minor comments:

Line 85 – to provide readers with greater understanding of the motivation of this experiment, could you indicate how often OSR in prairie voles fluctuates in natural populations? Is it likely that male prairie voles even experience female- and male-biased OSR in nature? If they do experience a male-biased OSR, couldn’t they just leave that area to find another?

Line 109 – are there any quantitative measures to show that the vegetation in the enclosures was the same?

Line 124 – could you give a few more details on the trapping schedule used here? Randomly distributed traps?

Line 127 – here the location of those animals that were not recovered should be provided. In addition, the authors should indicate what the vole OSR and density was in each enclosure at the end of the 4 week experiment.

Line 131 – are there data on how long it took to get the males from the lab to do this test?

Line 136 – is this thigmotaxis zone based upon some previous study? It seems arbitrary but this may just be my lack of familiarity here.

Line 153 – could you indicate 1) how many observers extracted behavioral data from the videos and 2) whether they were blind to the treatments?

Line 161- I’m not quite understanding why sex is in this model. Were females tested?

Line 162 – these should all be t-statistics since the numerator df are 1?

Line 166 – it would be helpful here to say something like “at the end of the trials, males from the MB found the platform x% faster than at the beginning whereas males in the FB enclosures ...” just to put some biological reality on the results.

Line 174 – this generalized linear model needs to report the dispersion parameter and whether or not it was overdispersed.

Line 202 – but isn't it difficult to test either of these hypotheses without a third treatment where OSR was equal? Similarly, the lack of multiple replicates for the OSR treatments is a major confound.

Reviewer: 4

Please see attached review file.

Author's Response to Decision Letter for (RSOS-190743.R0)

See Appendix B.

RSOS-190743.R1 (Revision)

Review form: Reviewer 1

Is the manuscript scientifically sound in its present form?

Yes

Are the interpretations and conclusions justified by the results?

No

Is the language acceptable?

Yes

Do you have any ethical concerns with this paper?

No

Have you any concerns about statistical analyses in this paper?

No

Recommendation?

Accept with minor revision (please list in comments)

Comments to the Author(s)

The authors have done a very good job at revising their manuscript in response to the substantive and lengthy suggestions made by reviewers. I applaud their dedication to detailing their revisions.

I feel that the manuscript is much improved, and the authors have satisfactorily addressed most of my comments. Specifically, the new additions to the text in the methods and discussion regarding the experimental design help the reader understand the potential limitations and complexity of the study. I think that the new sections in the discussion are particularly helpful additions.

However, I still have one issue that the authors should consider more fully. (All line references below refer to the tracked changes version.) I remain concerned that the FB group did not demonstrate learning in the water maze. I appreciate that the authors are clear with their goals in their response – they did not intend to test if the voles could reach a specific criterion or threshold. This is fine. However, they never demonstrated that the FB voles learned the maze and so their subsequent test (trial 11) of that group is irrelevant.

What I find most confusing here is the authors' claim that the FB group did learn the maze because the slope of the line in figure 3 is not zero and that there was a difference in performance between trials 1 and 10. I do not agree with this interpretation at all. Both of these criteria could potentially hold as a result of random variance if the voles indeed did not learn. The statistical analyses of those differences do not suggest learning. In fact, on line 262, the author state that the "FB males showed no difference in latencies across the learning trials" and provide statistics to support that point. Then, on line 264, the authors state that the FB group found the platform 17.5% faster, but this includes no statistical analysis. An anecdotal view of figure 3 suggests a great deal of overlap of the SEM, and so I doubt that this comparison is statistically different. So, in any other situation, one would interpret that the FB voles did not learn.

Although these voles are not in a situation of "profound memory or learning impairment" and the assumption is that they should be able to learn, this result still raises a red flag for me. On one hand, it is possible that the authors found a test with a complexity and difficulty level that exactly splits the two groups at their threshold (and thus resulting in one group that could learn and the other that could not). However, because the authors did not demonstrate that the FB group could learn, i.e., no positive control, they cannot dissociate whether the FB group's learning was not so good as the other group, if the Morris water maze was an inappropriate task for this group, or if something else might have been at play preventing them from performing in the maze that is independent of spatial memory abilities all together. Without this verification/control, one could speculate all sorts of things. Perhaps the males in that group were fatter (with more access to food resources with fewer individuals in the enclosure) and so could not swim as well.

My point here is not to suggest that the authors propose or address alternatives. I admit that the probability that my concern is creating a spurious effect is relatively small. Indeed, all of the arguments that the authors provide are valid arguments and the experiment itself helps reduce some of those concerns. However, from the perspective of experimental design, interpretation, and transparency, the authors really should address this caveat and admit that they cannot definitely rule out the possibility, however unlikely.

Line 389. I suggest that the authors are a bit more tentative here with the language and replace "indicate" with "are consistent with the hypothesis that".

Review form: Reviewer 3

Is the manuscript scientifically sound in its present form?

Yes

Are the interpretations and conclusions justified by the results?

Yes

Is the language acceptable?

Yes

Do you have any ethical concerns with this paper?

No

Have you any concerns about statistical analyses in this paper?

No

Recommendation?

Accept as is

Comments to the Author(s)

The authors have made numerous revisions to this manuscript after 4 thorough reviews. I'm very satisfied with the changes they have made and their detailed and thoughtful responses. My only comment here is that in some places (e.g., line 224) they are providing the numerators degrees of freedom for t statistics (=1) whereas others they are not (e.g., line 235). I think you do not need to indicate numerator degrees of freedom as 1 in these places such as line 224.

Decision letter (RSOS-190743.R1)

10-Oct-2019

Dear Dr Ophir:

On behalf of the Editors, I am pleased to inform you that your Manuscript RSOS-190743.R1 entitled "Social context alters spatial memory performance in free-living male prairie voles" has been accepted for publication in Royal Society Open Science subject to minor revision in accordance with the referee suggestions. Please find the referees' comments at the end of this email.

The reviewers and Subject Editor have recommended publication, but also suggest some minor revisions to your manuscript. Therefore, I invite you to respond to the comments and revise your manuscript.

- **Ethics statement**

- **Data accessibility**

It is a condition of publication that all supporting data are made available either as supplementary information or preferably in a suitable permanent repository. The data accessibility section should state where the article's supporting data can be accessed. This section should also include details, where possible of where to access other relevant research materials

such as statistical tools, protocols, software etc can be accessed. If the data has been deposited in an external repository this section should list the database, accession number and link to the DOI for all data from the article that has been made publicly available. Data sets that have been deposited in an external repository and have a DOI should also be appropriately cited in the manuscript and included in the reference list.

If you wish to submit your supporting data or code to Dryad (<http://datadryad.org/>), or modify your current submission to dryad, please use the following link:
<http://datadryad.org/submit?journalID=RSOS&manu=RSOS-190743.R1>

- **Competing interests**

- **Authors' contributions**

- **Acknowledgements**

- **Funding statement**

Because the schedule for publication is very tight, it is a condition of publication that you submit the revised version of your manuscript before 19-Oct-2019. Please note that the revision deadline will expire at 00.00am on this date. If you do not think you will be able to meet this date please let me know immediately.

Kind regards,

Lianne Parkhouse
Royal Society Open Science
openscience@royalsociety.org

on behalf of Dr Ryan Y Wong (Associate Editor) and Kevin Padian (Subject Editor)
openscience@royalsociety.org

Associate Editor Comments to Author (Dr Ryan Y Wong):

Dear Dr. Ophir,

Your revised manuscript has now been reviewed by two reviewers. Both reviewers also commented on the original submitted version. Both reviewers thought you adequately addressed most of the prior concerns. However, one reviewer noted a concern for the interpretation with the

FB group and the other noted an issue with consistency of reporting statistical information. I believe these can be addressed with a minor revision.

Reviewer comments to Author:

Reviewer: 1

Comments to the Author(s)

The authors have done a very good job at revising their manuscript in response to the substantive and lengthy suggestions made by reviewers. I applaud their dedication to detailing their revisions.

I feel that the manuscript is much improved, and the authors have satisfactorily addressed most of my comments. Specifically, the new additions to the text in the methods and discussion regarding the experimental design help the reader understand the potential limitations and complexity of the study. I think that the new sections in the discussion are particularly helpful additions.

However, I still have one issue that the authors should consider more fully. (All line references below refer to the tracked changes version.) I remain concerned that the FB group did not demonstrate learning in the water maze. I appreciate that the authors are clear with their goals in their response – they did not intend to test if the voles could reach a specific criterion or threshold. This is fine. However, they never demonstrated that the FB voles learned the maze and so their subsequent test (trial 11) of that group is irrelevant.

What I find most confusing here is the authors' claim that the FB group did learn the maze because the slope of the line in figure 3 is not zero and that there was a difference in performance between trials 1 and 10. I do not agree with this interpretation at all. Both of these criteria could potentially hold as a result of random variance if the voles indeed did not learn. The statistical analyses of those differences do not suggest learning. In fact, on line 262, the author state that the "FB males showed no difference in latencies across the learning trials" and provide statistics to support that point. Then, on line 264, the authors state that the FB group found the platform 17.5% faster, but this includes no statistical analysis. An anecdotal view of figure 3 suggests a great deal of overlap of the SEM, and so I doubt that this comparison is statistically different. So, in any other situation, one would interpret that the FB voles did not learn.

Although these voles are not in a situation of "profound memory or learning impairment" and the assumption is that they should be able to learn, this result still raises a red flag for me. On one hand, it is possible that the authors found a test with a complexity and difficulty level that exactly splits the two groups at their threshold (and thus resulting in one group that could learn and the other that could not). However, because the authors did not demonstrate that the FB group could learn, i.e., no positive control, they cannot dissociate whether the FB group's learning was not so good as the other group, if the Morris water maze was an inappropriate task for this group, or if something else might have been at play preventing them from performing in the maze that is independent of spatial memory abilities all together. Without this verification/control, one could speculate all sorts of things. Perhaps the males in that group were fatter (with more access to food resources with fewer individuals in the enclosure) and so could not swim as well.

My point here is not to suggest that the authors propose or address alternatives. I admit that the probability that my concern is creating a spurious effect is relatively small. Indeed, all of the arguments that the authors provide are valid arguments and the experiment itself helps reduce some of those concerns. However, from the perspective of experimental design, interpretation,

and transparency, the authors really should address this caveat and admit that they cannot definitely rule out the possibility, however unlikely.

Line 389. I suggest that the authors are a bit more tentative here with the language and replace "indicate" with "are consistent with the hypothesis that".

Reviewer: 3

Comments to the Author(s)

The authors have made numerous revisions to this manuscript after 4 thorough reviews. I'm very satisfied with the changes they have made and their detailed and thoughtful responses. My only comment here is that in some places (e.g., line 224) they are providing the numerators degrees of freedom for t statistics (=1) whereas others they are not (e.g., line 235). I think you do not need to indicate numerator degrees of freedom as 1 in these places such as line 224.

Author's Response to Decision Letter for (RSOS-190743.R1)

See Appendix C.

Decision letter (RSOS-190743.R2)

23-Oct-2019

Dear Dr Ophir,

I am pleased to inform you that your manuscript entitled "Social context alters spatial memory performance in free-living male prairie voles" is now accepted for publication in Royal Society Open Science.

Best regards,

on behalf of Dr Ryan Y Wong (Associate Editor) and Kevin Padian (Subject Editor)
openscience@royalsociety.org

Associate Editor Comments to Author (Dr Ryan Y Wong):

Dr. Ophir,

I am pleased to announce that I am recommending the manuscript be Accepted.

Appendix A

RSOS - Social context alters spatial memory performance in freeliving male prairie voles (*Microtus ochrogaster*)

This is an enjoyable paper that sets out to measure the effect of the social environment on spatial learning and memory performances. Specifically, the paper explores the effects of two different types of competition (contest versus scramble) on male spatial performances, by comparing two prairie vole groups, each exposed to a different sex-ratio treatment. The paper reports that males living in the male-biased sex ratio treatment outperformed males that have previously lived in the female-biased sex ratio treatment. The paper concludes that intensity of contest competition (male-male competition) causes superior spatial ability performances, whereas scramble competition (fewer females) does not.

I think this paper holds much promise and will be of interest to readers of RSOS. However, I had some concerns regarding justification of the experimental design and the interpretation of the results. It is unclear that the female-biased sex ratio treatment represents scramble competition and the male-biased treatment does not? The definition given in the introduction states that scramble competition “*involves males searching for females, particularly when the number of available females is limited.*” – Why are females *more* limited in the female-biased population compared to the male-biased? It is also not clear that the data support or allow you to distinguish between the two types of competition with your sex ratio treatments. Additionally, the interpretations of findings require toning down. Specifically, because the number of females was kept constant across the treatments and was not varied experimentally, the paper is not able to make inferences about the effect of searching for females on male spatial performances. The final concern is that the treatments differ in overall group size and the number of males – both of which, may be the causal factor in effecting spatial performances and this needs to be acknowledged.

The specific statistical analyses conducted are unclear and the paper would benefit from a dedicated section about the analyses for each response variable and the tests used with justification for each. There is also confusion in the results section regarding which fixed effects were included in analyses. Specifically, it is unclear why sex was included as a fixed effect if only males were included in analyses?

Please see comments below regarding specific areas that need further clarity.

Abstract

It is unclear from the abstract how the different sex-ratio treatments relate to the two types of competition.

Line 24: “Equally complex spatial context” – of the task or the environment?

Introduction

I struggled a little bit with the opening paragraph, there are some nice examples and it does a good job of drawing the reader in, but it would benefit from a bit of rephrasing. Please see suggestions below:

Line 32: “*Navigating the world...*” use ‘environment’ instead of ‘world’, it is then consistent with the rest of the MS.

Line 33: Change “...*spatial locations of contexts, presumably that lead to salient features of the world (like food sources or familiar territories)*” to “...*the location of salient features in the environment (e.g. food sources or territories)*.”

Lines 35-36: “*although some recent work has begun to consider these questions.*” – such as?

Line 44: use ‘Similarly,’ instead of ‘In another example,’

Line 46: In what way is this an example of episodic-like memory? Is this ‘what, **when** and where’ important to your hypotheses or the story of the paper? If not, I wouldn’t use this term because it requires explanation, instead, just focus on Foley et al.’s findings that contribute to your questions/story.

Line 54: References and some finessing of the definitions of the two types of competition would be helpful here. How does female availability relate to contest competition and is it important to the definition? Are there situations in which you can have contest competition and no scramble competition?

Line 58: resources that attract who? The males or the females? Please rephrase.

Line 60: This line states that intensity of intraspecific competition (male-male competition) influences male-male competition. Please rephrase.

Lines 70-72: “*Taken together, variation in OSR can establish social contexts in which variation in behavioural tactics (such as how to approach intraspecific competition) reflect solutions to those immediate contextual challenges*” – it is unclear what it meant by “*how to approach intraspecific competition*”, what behavioural tactics exactly are you referring to?

It would be nice to know a bit more about the natural history of the voles. How are females distributed in the environment? How does a female choose a territory and then how does she choose a male? What are the average male and female home range size? Is there much reproductive skew in natural populations? What are the natural levels of competition between males and what does this consist of? Physical contests? Scent marking? Female mate-guarding?

Methods

Line 91: “...*unrelated pairs of F1 or wild-caught...*” – how many did you have of each? Or do you mean that F1’s are your wild-caught individuals?

Line 92: change to ‘old’

Line 94: Please specify what dimensions these measurements represent

Line 97: Cool. I know nothing about ear-tagging voles! What materials do you use? Are they numbered/coloured/symbols? Please elaborate.

Line 109: delete “...*directly comparable and...*”

Lines 113 – 117: I appreciate that the male-female ratio’s were kept the same, but reversed, across the treatments. However, I am unclear why it was more important to do this than standardising overall group size? There is now an added confound, as group size (with no mating competition) has been shown to have a positive causal effect on cognitive performances in an avian species (Langley

et al. (2018). *Animal Behaviour*, 142, 87–93). Similar processes could also affect the voles. It therefore makes it difficult to conclude that it was a difference between the treatments in male-male competition that caused the differences in spatial ability. It could simply be caused by a difference in the overall number of individuals.

Line 118-121: This is a bit unclear. Are you arguing that female distribution is relatively uniform across the two treatments and therefore the only difference is male-male competition and the number of females? It could still be a function of density – I'm not sure the data allows you to rule that out as a possible explanation.

Line 126-127: please give the survivors for each sex within each treatment.

The American spelling of 'behavioural' is used throughout the MS. I'm unsure what the journal prefers?

Line 131 – were subjects tested individually? Please be clear.

Lines 133-141: It would be helpful to describe the apparatus first and what the animals were required to do and then describe your zones and their measurements.

Line 146: Were all 10 learning trials done on the capture day? What was the inter-trial interval? This is particularly important for the memory trial – how long after trial 10 was the memory trial conducted? Ensure methods are reproducible for the reader.

Figure 2: I found the arrows on the diagram a bit confusing; it looks as if when you get to thigmotaxis zone near the platform, they overlap, but I realise these arrows are not to scale. Perhaps using dashed lines to draw the zones (approximately) over the photo would be helpful.

Figure 4: Is this the spatial **memory** task? Please include in figure legend.

Results

Throughout the results 2dp and 3dp have been used. There are also some formatting differences, such as a space between df in some cases but not others; sometimes there are commas between each test statistic and sometimes not. Please be consistent.

The paper requires elaboration on the statistical analyses used and I would suggest including this as a separate section at the end of your methods section and to walk the reader through which analysis was conducted for each question. That way, discussion of methods can be removed from the results so that **only** results are stated here. In the methods section it would be useful to include the following: What software was used? Was this a Gaussian distributed mixed model? Were your model residuals normally distributed? If not, you may need to think about the appropriate error structure to use as latency data is characteristically poisson distributed. Please justify your choice of model for **all** questions. How were your F statistics generated? Are they from comparisons of model deviances? Any model simplification? Please explain.

Line 159: This is the first mention that females were tested – your hypotheses are purely based on male cognition in relation to differences in male-male competition/locating females. So it is unclear why 'sex' is included as a fixed effect? The paper has not explained why females are included in analyses and what differences we'd expect to see.

Line 161: How have you “...controlled for...” swim velocity and thigmotaxis? Please explain, were they included as a fixed effects?

Line 164: In the description of the analyses above there is only mention of a sex * trial interaction...not treatment * trial. Please ensure the results match what is described above.

Line 177: first mention of proximity to the goal being narrowed. Please explain.

Lines 179-181: This belongs in methods

Discussion

Lines 195-196: “...males must rely on their spatial memory to locate potential mates, particularly when females outnumber males and conditions are ripe for intense scramble competition...” – the definition in the introduction states that scramble competition occurs when females are limited. In this case, females are not limited. Please explain.

The number of females is identical across the treatments (scramble competition has been ‘standardised’), yet contest competition has been varied. Therefore, the data justify the discussion that contest competition may be driving differences in spatial performances but the data does not allow you to comment on the effect of scramble competition because this has not been varied. Some clarity is needed throughout the discussion and it feels like there is new justification of hypotheses in the opening of the discussion that really should have been detailed in the introduction and then summarised here.

It would be nice to open the discussion with a summary of your findings, then go on to explain how these findings support or refute your hypotheses (which are detailed in your intro) and fit into existing literature.

Line 197: It is correctly stated that the two potential mechanisms for a difference in spatial ability are non-mutually exclusive. It is unclear how the findings allow you to separate them. Please explain.

Lines 200-202: “Specifically, we found that males exposed to a male-biased environment in which competition for living in a social context in which males outnumbered females out-performed males living in a social context in which females outnumbered males.” – this sentence is a lot to digest, a few commas may help to break it up and it’ll be easier to process. It would also be good to include how they outperformed the other males, i.e. spatial learning and memory tasks.

Line 205: “This result is consistent with other work that found male prairie voles attend to the social identity of males more closely than females (Zheng et al. 2013)” – I’m not sure on the relevance of the comparison here, in what way do males attend more to males than they do to females? Please explain.

Line 214: Please include “*spatial learning and memory*” throughout the manuscript (unless of course discussing the individual measures) or refer to them collectively as “*spatial ability*”

Line 217: Cognitive or behavioural ‘flexibility’ (more commonly measured using reversal learning or inhibitory control tasks) was not assessed in the paper, so I would be careful with the wording here.

Line 219: “*The form that such a response should take...*” – What response is this? Response to social composition/sexual competition or their cognitive performance? Please explain.

Line 222: delete “*...(and perhaps should)...*”

Line 231: just spatial memory or spatial learning too? The opening sentence discusses memory specifically but the examples given regard both learning and memory. Please be clear.

Line 242: “*...the performance of all our subjects should have been equally strong.*” – as strong as what? Each other? Do you mean that there would have been no difference in spatial learning and memory between the treatments because all other factors were identical and only the social composition differed?

Line 245: delete “*...(not spatial)...*”

Line 248: In what way are these environments ‘spatially complex’? – There is not much description in the methods of this complexity. It is not overly important to state this anyway, as this is not something that was varied experimentally, it is simply important to highlight that the treatments were spatially identical and varied only in their social composition.

Lines 249-250: “*To our knowledge, this is the first demonstration that social context, while holding spatial complexity constant, impacts cognitive functions related to spatial information.*” – this is not entirely true, social context (in terms of group size and individual social interactions) has been shown to influence spatial learning in other taxa. Please see:

- Langley et al. (2018). Individuals in larger groups are more successful on spatial discrimination tasks. *Animal Behaviour*, 142, 87–93.
- Barnard, C. J., & Luo, N. (2002). Acquisition of dominance status affects maze learning in mice. *Behavioural Processes*, 60, 53–59. doi: 10.1016/S0376-6357(02)00121-3

Line 252: The spatial environment was not manipulated so I would remove “*spatially complex*” here.

Appendix B

Dear Drs. Wong and Padian

Thank you for the opportunity to revise our manuscript: *Social context alters spatial memory performance in free-living male prairie voles (Microtus ochrogaster)* (Manuscript ID RSOS-190743). We have worked to carefully address the thoughtful and constructive feedback we received from the four reviewers. Most notably, we addressed the issue that all four reviewers raised regarding the potential confound of population density and sex ratio in our responses below and in the manuscript. Of course, for all of the collectively extensive comments we received we have modified our manuscript and/or addressed them in our responses below and we hope our responses are satisfactory to all. Below, our responses are indicated by blue italic font.

Reviewer 1

This is a relatively short and well written paper comparing the spatial memory of male voles that had experience in two different experimental sex-ratio conditions. This study builds off of the vast amount of work suggesting that the physical environment has an effect on memory, both at the ontogenetic and phylogenetic levels. One of the overarching goals of the paper was to demonstrate that the social environment also plays a role in the creation/manipulation of memory. The manuscript is clear and is for the most part fairly straightforward. The topic is current, very interesting, and has the potential to make an important impact on the field. We certainly do need to consider the role of both the social and physical environments when studying the formation and maintenance of memory.

1. Although I really like the paper, the main weakness, in my opinion, is in its interpretation of the results given the nature of the experimental design. The authors manipulated the sex ratio of the experimental groups by keeping the number of females constant and changing the number of males, thereby changing the competition among males for access to females. [In doing so, they also manipulated vole density and thereby changed the competition among all of the voles for resources]. Although the authors acknowledge this confound, they spend virtually no additional time discussing it or addressing its possible effect on their conclusions. The problem here is that the directions of the effect on competition are equal for both the sex competition and general resource competition hypotheses. Therefore, the authors cannot conclude that the change in memory in the males was necessarily a function of the change in competition for females.

I do not think that this confound is a deal-breaker, per se. The authors have important evidence that memory capabilities can be affected relatively quickly by changes in social conditions (and that the memory for one situation can apply to other situations, so based on ability to learn/form memory rather than simply the amount that has been learned). However, the authors need to spend much more time considering alternative hypotheses, especially in the discussion. Given the experiment, I see a place for both ideas (sex ratio and general resource competition) being interpreted as the effect of the social environment on memory.

We thank the reviewer for raising this important and fair point. We had spent a lot of time considering this issue before submitting our paper and we acknowledge we should have spent much more time explaining how resource competition relates to male mating behavior and competition.

To address the criticism here, and as we explain in the revised manuscript (Methods and Discussion), we carefully decided to hold female numbers constant and manipulate the relative number of males to females deliberately based on both empirical work in voles and theoretical work on mating systems. Classic mating system theory outlined by Emlen and Oring (1977) argued that the potential for polygyny is directly impacted by the distribution of females in space. This has been strongly supported by other theoretical work (e.g., Shuster & Wade 2003). This theoretical work has been the foundation for the idea that 'females track the environment, and males track females'. This important premise was paired with the results from Myllymäki (1977) which demonstrated that female field vole distribution and home range size are impacted by food distribution (i.e., how uniform or stochastically distributed it is), and not the population density. Our experiment was designed to ask if males' (memory-based) behavior would be altered by the social landscape. Altering the absolute number of females would have had the consequence of changing the amount of space each female was able to occupy, which would have introduced an additional but different confound (i.e., fewer females occupying more space vs more females occupying less space). Therefore, we purposely held constant the number of females within a fixed area of space (the enclosures were identically sized) so that we could evaluate how the impact of male-to-female ratio would alter behavior in contexts in which the same number of females were distributed over the same amount of space. With the knowledge (theoretical and empirical) that females' distributions would not be impacted by population density, we felt confident that any changes in behavior would be more likely attributable to male response to the sex ratio in relation to the fixed number of females per unit space. Again, we prioritized controlling for the potential confound of altered females per unit space (i.e. holding an independent variable constant) over the potential (but unlikely) confound of population density because our experiment focused on space use issues as our dependent variable.

We acknowledge the possibility that male behavior could have been impacted by population density in addition to or instead of the sex ratio. But we believe the available work referred to above strongly supports the interpretation that males reacted to the availability of females relative to males, independently of the total population density. We have now explicitly addressed this issue in the discussion (and elaborated on the justification for our approach in the methods), and we have acknowledged that population density cannot be ruled out, however probable or improbable this possibility might be. Either way, as the reviewer stated, these data are particularly interesting and important, and by acknowledging the potential population density confound, we better enable the reader to reach their own conclusion, even if it is inconsistent with our own preferred interpretation. To this end, we agree and thank the reviewer for helping us directly address this important point.

In addition, some discussion about the possible mechanisms for this effect (e.g., change in CORT, neurogenesis) could help contextualize the effect, which might be helpful for generating future hypotheses and predictions.

If the authors have any evidence of the memory abilities of the females from this experiment or other mechanistic data to support the changes in memory, those would be nice to include (or at least reference) here.

Unfortunately we did not test the memory abilities of females and we also did not collect any physiological measures/samples from males that would provide mechanistic explanations for our behavioral findings. However, we have spent time thinking about the potential mechanisms that might drive this alteration in spatial memory. We were reluctant to discuss these things in this manuscript (like CORT or neurogenesis) because we did not investigate them, because we were concerned these would be overly speculative, and because we wanted to keep the focus

on the social mechanisms rather than the neural or hormonal mechanisms that might have caused the memory differences. However, in response to this question and to Reviewer 2 (See below) we have very much made a passing reference to some of the many physiological mechanisms that could have accounted for the behavioral outcomes, including some of what the Reviewer mentioned.

2. Methods. The authors should include some additional information about the transfer of the voles from Illinois to Oklahoma. This should be done for two reasons. First, it was not particularly clear that they were from two different areas. I missed this the first few times. So, I suggest making this point specific. Second, a note about the similarity and/or differences in habitat between the sites is important. This is a minor point, but something that would be appreciated for clarity.

We agree that this added clarity is helpful for readers. The breeders in our colony were originally trapped in Urbana-Champaign, Illinois or the first-generation offspring of these wild-caught animals. Animals used in this experiment were the offspring (i.e. F1 or F2) of these breeders and were all born and raised in our breeding colony located in Oklahoma. We have now clarified this.

The habitat across the prairie vole distribution is highly variable (e.g. see Mabry et al. 2011 Animal Behaviour) but both the original trap site and our experimental site fall within the known prairie vole natural distribution. We now also make this point clearly in the methods. We have reported that testing different populations of animals (those trapped in IL or Tennessee) in the same location (TN) produced no differences in behavior (lab or field), and demonstrated no differences in the brain or genetic metrics we investigated (Ophir et al. 2007 J Mam) indicating that regardless of the original population from which animals were trapped, they behave the same in the same environments.

3. Lines 118. This point relates in part to point 1 above. The argument here suggests that female home range size is not affected by female density. I find this hard to accept from an ecological perspective. Perhaps at low to moderate densities, home range size would not be affected; however, at very high densities there would be some effect, if for no other reason than access to resources. This should be clarified or nuanced. Moreover, within the context of the experiment and densities therein, this point may stand for females, but the females are not the subjects of the study. What about males? How are male home ranges expected to change with overall changing vole densities?

We have modified this section of text as we discussed above.

4. Figure 3. It seems that the males in the Female-Biased group did not learn the water maze. The time to solve the maze did not decrease over the course of the trials. This is a problem. It is odd that they could not or did not learn at all. Is there some other explanation for why this inability to learn was occurring? I buy the idea that those males in the Male-Biased group would need to use and hone their memory and so might do better on subsequent memory tests. However, I find it suspicious that the Female-Biased group showed no learning at all. In addition, why did the authors stop at 10 trials? Why test the Female-Biased group when they were not at criterion?

According to our statistical model, there was a sex bias by trial interaction. This was mainly driven by the male-biased group but not the female biased group. That is to say that the female-biased group did not show a significant change in learning per trial. However, we overstated our

finding to conclude that females did not learn at all and we thank the reviewer for highlighting this mistake. We have corrected this in the manuscript.

We note that another way to visualize this is to compare the slopes in Figure 3. The female-biased slope is close to horizontal, but is not zero indicating that learning occurred, but at a relatively slower rate. Furthermore, there is an overall difference between trial 1 and trial 10 for FB males, again indicating learning occurred.

We have clarified the description of the procedure in Methods. We stopped at 10 trials because we were following conventional Morris water maze procedure, (e.g., Rice et al. 2017). Our assessment of spatial memory was not aimed at evaluating if animals could reach a particular criterion or threshold of learning, but rather to assess the rate and capture the process of learning and memory, on the assumption that all voles have the capacity to learn this task at some point (i.e., there was not expectation that animals should have profound learning or memory impairment typical in biomedical studies).

5. Figure 4. I am having difficulty resolving the lack of learning in the Female-Biased group (from Figure 3) with the very small differences in time spent in the quadrant/platform areas indicative of spatial memory/learning of this figure. The mean difference between the groups is only about 5 seconds of time in (Fig 4a) and 2 visits to (Fig 4b) the correct quadrant and 1.5 seconds near (Fig 4c) and 2 visits to (Fig 4c) the platform area. These differences seem tiny relative to the context of 120 seconds (max time) in the maze and the average of 50 seconds difference in time to find the platform (Figure 3).

As noted above, the FB males did learn, just at a slower rate. We also want to note that the maximum time in the memory trial (Trial 11) was out of 60 seconds, not 120. However to your larger point, we acknowledge that the differences appear to be relatively small, but we also emphasize that small effects can be substantial (e.g., Rice et al. 2017). Furthermore, MB males exhibited better accuracy in locating exactly where the platform was, as demonstrated in the proximity to the platform metric during the memory trial (a difference of approx. 13cm). We have now added this figure to help convey the strength of the overall differences. To this end, we have also clarified that the memory differences highlight accuracy and precision. Additionally, we have included effect sizes for all reported measures.

Reviewer 2

Adult prairie voles were released into semi-natural enclosures with either a male-biased (MB) or female-biased (FB) sex ratio (SR) for a period of four weeks, after which they were re-captured and males were tested in the Morris water maze to examine spatial learning and memory. After the four-week treatment MB males showed significant improvement in latency to locate the hidden platform over successive trials in the water maze while FB males did not; MB males also visited more often and spent more time in the correct escape zone during the probe trial. This experiment is simple and has provided clear, consistent results, but has not properly tested what the authors claim to have tested. While sex ratio is the treatment effect of interest, it is not sex-ratio per se, but rather the effects of male density that are tested in this experiment. As such, I think the background, objectives, hypotheses, and discussion could use substantial revision.

We deeply appreciate this point. As we indicated in our response to Reviewer 1, who also raised this, we had considered this when we were designing the experiment, and chose to design our experiment this way based on both theoretical and empirical support that population

density is highly unlikely to influence the results, whereas altering the number of females per unity space would pose a more serious confound to our design. We admit that we should have been much better about explaining the justification for this design and providing a discussion of this at the end of the paper. We have now done both. Please see our response to Reviewer 1, Point 1 above for additional details or Methods and Discussion in the manuscript.

Missing from the introduction are the reasons why SR (or rather, male density) might influence male cognitive abilities after four weeks. While it is clear that natural selection can shape spatial learning and memory, it is not clear why the environment an adult experience would influence their spatial abilities.

We neglected to mention the crucial link in our introduction between memory and environment and thank the reviewer for raising our awareness of this shortcoming. Indeed, there is a rich literature demonstrating that spatial cognition is plastic and can be shaped by the immediate demands of the environment during adulthood. We have added this key point in the introduction and discussion.

The study on male meadow voles suggests intraspecific variation in a congener but does not indicate why this phenotypic link exists. Some details are provided in the discussion (L214-229) that could be incorporated into the introduction. As is, the theory supporting the hypothesis that SR influences an individual's spatial cognition are not clearly laid out. What mechanism could be at work over a four-week period?

We have added information about plasticity and spatial flexibility in the introduction. We have been careful not to speculate too much about mechanism because we did not collect any physiological data that would support inferences about any particular mechanism. However, there is substantial literature, specifically focused on the hippocampus, concerning the various ways in which spatial plasticity in the brain exists on a rapid time scale. We have included this in the discussion to give more support to our hypothesis and thank the reviewer for noting this.

As mentioned, the hypothesis tested is not quite what is stated: while both scramble and interference competition were mentioned, the design of this experiment only allows for examination of the effects of interference competition on male cognitive abilities. Because there are 12 females in each treatment there is no way to compare spatial abilities relative to differences in scramble competition. Rather, this experiment is only testing the effects of male density, and if differences in male density and the hypothesized competition inherent in lower vs. higher male density affect male's spatial learning and memory abilities.

We respectfully disagree with the reviewer's interpretation of our design. In fact, we believe that what the reviewer has described is exactly what we intended to test, because altering the relative density of males to females is the central idea underlying an altered sex ratio. This could be accomplished in at least three ways: 1) change the number of males while holding the number of females constant, 2) changing the number of females while holding the number of males constant, or 3) changing the number of males and females relative to each other. For the reasons explained above we chose option 1.

In regards to the results and discussion, it is again important to note that the effects of male-biased sex ratio cannot be separated from the effects of density. To separate the effects of sex ratio and density, enclosures with even sex ratios but density equivalent to the enclosures used in this experiment would have provided crucial, proper controls. This would allow for the effects of SR itself to be examined.

Please see our response to this critique above.

Specific comments:

L15- 'time' how is this related to spatial memory? This would be time/place learning. You can have spatial learning without a time component. Please be specific. Did you test time learning?

Good points, we simply meant in real-time. We have eliminated the word time to avoid confusion.

L23-4- Why are there difference, then, between the MB and FB conditions? What is driving the differences? I would remove this statement because we do not know what 'complex' means for these animals.

We have clarified this in the manuscript. This statement is important because being in a semi-natural field enclosure is arguably more complex than the standard shoe box cage where they are reared. However, both the FB and MB group experienced this spatially complex environment in the field enclosure. But it was the social context, i.e. sex ratio, within each enclosure that produced the behavioral difference.

L29- change cognitive "feature" to "ability"

Done

L32- navigation does not necessarily require spatial representations or "mental maps", e.g. step-counting

We have made a clarification here to exclude procedural methods

L46- 'episodic-like memory' What is this and why is it important to the current experiment?

Removed 'episodic-like memory'

L50- use 'performance' instead of 'capacity'

Done

L52- confusing as I thought the authors have been trying to articulate that mating behaviour is an important factor in determining spatial memory?!?

Changed 'determined' to 'influences', because we are acknowledging that spatial memory and mating behavior are related and bi-directional

L111-Could they hear vocalizations from the other enclosure - what potential effect could this have?

No, it is extremely unlikely that the voles were able hear vocalizations from the other enclosure; long distance auditory communication in rodents is highly uncommon, especially at the distances between (or even within) our enclosures.

L118- 1977 reference – has this statement regarding distribution/home range size been verified in prairie voles? Density does seem to at least affect social associations (see Streatfeild et al. 2011 Animal Behaviour).

It has not been verified experimentally in prairie voles, however the field vole is a very close relative. Streatfeild et al. (2011) did address how density impacts social association (or social networks) in prairie voles, however as those authors noted 1) this is not the same as social distribution in space, and 2) they were unable to disentangle social associations from the distribution of resources (i.e., vegetation). To this end, we believe Streatfeild et al. (2011), reinforces the main claim we are highlighting made by Myllymäki (1977), which is that distribution of vegetation is the main factor that influences distribution. We also note that our vegetation was relatively uniform across and between enclosures. We therefore feel as confident as we possibly can that effects we report can be attributed to the ratio of males relative to females and not population density. But see our additional discussion of this in the Discussion.

L124- Please describe how animals were trapped. The trapping period to being animals back to the lab was up to 25% of the duration of the treatments. How could this affect your results? Was date of trap (and resultant SR [and density]) accounted for in the analyses?

We have added more details about our trapping procedure (see Methods). To clarify, the majority of trapping was done within a day or two of the trapping period (only 5 days). As a result we did not account for date of trap in our model because of the quick trap turn around, and we believe that a difference of one or two days, and the resultant sex ratio within those 24-48 hours would be negligible compared to the 4 weeks maintained at our desired context ratios.

L183-5 This does not belong in the results

We respectively disagree and feel a simple summary statement of the results improves the readability of the paper. For this reason, and as a matter of stylistic preference, we prefer to keep it in.

L212- the adaptive specialization hypothesis does not apply to individual cognitive plasticity does it?

Yes, we agree. This applies at the level of population/group, thus groups within the same species facing different ecological demands (various contexts) could arrive at different cognitive solutions.

L215- also see Buchanan et al. 2013 Trends in Ecology and Evolution

Citation added

L251- Conclusion paragraph highly repetitive with previous paragraph.

Eliminated redundancy

Discussion- Did you measure behaviour? Do you have any idea what the males in the different treatments were doing?

Unfortunately, we did not directly measure behavior of males while they were inside the enclosure (and we deeply regret not having done this). The best we can do is infer the type of intraspecific competition faced in each enclosure based on the previous work that did measure aspects of male behavior.

Fig. 3 & 4- perhaps add sample sizes to figure, either in figure captions or in figures themselves

sample size added to figure captions

Reviewer: 3

Vole spatial memory review

This is a very interesting study that examines how social context experienced by male prairie voles influences their spatial memory. Animals experience fluctuations in operational sex ratio (the ratio of males to females in a population) and this can influence males in the population by affecting the number of competitors that are trying to find females or the number of other males that may interfere with their mating behaviour. The authors put male prairie voles into one enclosure with a female-biased operational sex ratio (40% males) and other voles in a separate enclosure with a male-biased operational sex ratio (60% males). At the end of the 4 week period, they tested the spatial memory performance of males using a Morris water maze. The authors show that male voles from the enclosure with the male-biased operational sex ratio had higher spatial memory than those from the enclosure with a female-biased operational sex ratio.

These results are very interesting as the authors note that such studies investigating the role of the social environment and how it might influence changes in spatial memory are quite rare (especially in animals living in natural or semi-natural environments). The manuscript is clear and well-presented and I have only a few questions about the statistical analyses (listed below). I do have some questions and request for elaboration on a few points below. Overall, I think this is a very interesting study but the authors just need to address a few additional points and discuss some of the limitations.

We thank the reviewer for their positive and encouraging comments regarding our manuscript.

Major comments:

The authors present results from one enclosure where voles were housed at a female-biased OSR and one enclosure where the voles were housed at a male-biased enclosure. Thus, there are some limitations that might be mentioned in the Discussion such as 1) they effectively have one statistical replicate of each treatment but are using individuals within each of the enclosures as their unit of replication and 2) they do not have data on male spatial memory performance in enclosures with an equal OSR.

Our response to these two points are detailed in the two sections immediately below.

I think it would be helpful here to make note of the unit of replication here and its limitations and also make clear to readers why they think the different OSR in the one enclosure is the actual factor causing the difference and not some other feature that differs between the two enclosures.

The reviewer raises a relevant point that the enclosure can be considered the sample (i.e., N=1) rather than the individuals within the enclosure and that other enclosure-based factors could have contributed to the results we reported. We note that even when multiple replicates are run, data such as these are often analyzed, treating the individuals as the sample, but using a nested design to control for enclosure effects. The limitations of doing what we did, largely boil down to potentially reduced statistical power, and possibly semi-independent data within each enclosure. With respect to reduced statistical power, we agree that adding replicates would have certainly improved statistical power, but if our results are sufficiently robust enough to reach statistical significance with relatively low power, then this only reinforces these results are compelling and meaningful.

The animals acting within each enclosure very well interacted with each other (that was the intended design). Research on wild populations rarely receive the same criticism (rightly or wrongly) and the only reason this seems particularly relevant in our case is because we established the composition of the enclosure, whereas studies on natural populations do not have that degree of control. We view this as a strength of our study design, because we were able to create populations of individuals that were randomly chosen from the same overall stock of animals (animals living in each enclosure were drawn from the same breeding colony and breeders). Therefore, each animal was introduced into a socially unique context (male- or female-based ratios) with identical pre-existing social, genetic, and developmental histories and thus began the experiment on the identical footing. These points are now included in the Methods. We believe this dramatically reduces any opportunity for sampling biases that multiple replicates are used to avoid.

With respect to the potential for enclosure effects, the vegetation and ecological and physical landscape was essentially the same (something we clearly note in the revised manuscript now. We also note here that in other (currently unpublished) experiments in which we have run multiple replicates in these enclosures, we never found enclosure effects that altered any behaviors we measured (including things like the ways animals used space, interacted with other conspecifics within the enclosure, or sired offspring). We also found that the physical condition of the animals living in the two enclosures was identical between enclosures. All of this supports the conclusion that the enclosures were uniformly comparable and similar.

To sum up, the enclosures were identical in all respects other than social composition, the animals we used were randomly selected from the same sample of animals that had the same life history experiences, and we found statistically robust effects even in the presence of potentially lower power than what multiple replicates offer. These points collectively suggest that the results we report are not a function of sampling bias or enclosure effects, and represent differences attributable to the social context we established.

Second, I think it would be helpful if the authors indicated what they would expect regarding male spatial memory performance in enclosures with equal OSR (perhaps in the Introduction and acknowledge that this wasn't performed and why). This comes up again in the Discussion (e.g., lines 203-205) where the authors interpret their results but it is difficult to do so without a third treatment of equal OSR.

We did not run an enclosure with an equal sex ratio, because this would have certainly masked any effects of how the social context and dynamics between males and other males, or males and other females impacts spatial memory. As with any behavior, there is individual variation in a sample, and (based on the data we have now as a result of the study) animals that are more predisposed to the presence of other males would have presumably been better at spatial

memory than those that are less sensitive to the presence of males or more sensitive to the presence of females. Either way, the data set would have produced results that capture something in between. This would have effectively been a very noisy data set that would have been difficult to interpret and might have impaired our ability to make sense of the data from the two groups we did run. Because we wanted to test mutually exclusive hypotheses on how the social context might impact cognition, we chose to set up two extreme versions of social contexts that enabled us to test each hypothesis rather than also including a middle ground compromise group that we think would have been relatively uninformative and possibly distracting from addressing the question we aimed to test.

A third limitation from the study design is that the authors had two treatments that varied the proportion of males in each enclosure but the density of voles also differed. Specifically, the male-biased OSR enclosure was 18 males and 12 females (60% males) and the female-biased OSR was 8 males and 12 females (40% males). It would be useful for the authors to note whether these treatments are within the natural range of variation in OSR – are they similar to what is found in nature?

Thank you for raising this question. Yes, the OSR used in the experiment was within the natural range of variation of prairie voles, we have added this citation (Getz et al. 1987) to the manuscript in the Methods.

Second, this creates a confound with density where vole density in the male-biased OSR is 375 voles/ha and 250 voles/ha in the female-biased enclosure (so 50% higher in the male-biased enclosure). Similar to the above, the authors should note the densities here and whether these are within the natural range of variation. Although the authors acknowledge that this difference in density complicates the analyses in the Methods section (line 114), I think they need to provide greater discussion as it is not clear to me how they can differentiate the effect of density vs. OSR. One thought is that male home range sizes may vary as a function of density (is this known in this species?) and males in the male-biased OSR with higher density might have lower home range sizes. We therefore might expect that males in the male-biased OSR would have poorer spatial memory abilities and thus the results presented here (that males in the male-biased OSR actually have higher spatial memory) do not support that prediction.

We obviously overestimated how well we dealt with the issue of population density in our original draft and thank the reviewer for helping us realize this. Indeed Reviewers 1 and 2 also noted this issue. Rather than recap our response at the risk of being redundant, we refer the reviewer to our response to Reviewer 1 (Point 1), where we tried to clarify this issue better. Briefly, we did consider this when we were designing the experiment, and chose to design our experiment this way based on both theoretical and empirical support that population density is highly unlikely to influence the results, whereas altering the number of females per unity space would pose a more serious confound to our design. We admit that we should have been much better about explaining the justification for this design and providing a discussion of this at the end of the paper. We have now done both in the revised manuscript (Methods and Discussion).

It is fascinating that in only 4 weeks of exposure to a male-biased population sex ratio, male voles exhibited greater spatial memory performance. However, one of the notable missing pieces of data is a “before treatment” evaluation of their spatial memory performance. This should be discussed.

We thank the reviewer for raising this point – and agree that this effect is quite fascinating. Unfortunately, we have no information about the baseline levels of spatial memory prior to treatment. We have acknowledged this point in the manuscript.

I'd like the authors to provide a greater context on their results. How quickly can spatial memory change in response to adjustments in the social context in other lab studies? This is not a major criticism. I just find the 4 week time period to be intriguing and would like to know more and the authors could help readers here by providing some context.

We wish we could provide more data on this, but to our knowledge, our study is the first to demonstrate in a field study that the social context can alter spatial cognition. We do discuss the degree of plasticity in general in spatial memory, but beyond this, there is not much out there that speaks to this point. This is one of the reasons we think this study is particularly exciting and important and we hope the reviewers, editors, and readers of our paper agree.

The authors might broaden the manuscript by focusing on some of the underlying machinery that drives the generation and storage of these “mental maps” that are influenced by social context. Perhaps in the Introduction it would be helpful to give readers an idea of what the current state of our knowledge is regarding how these maps are created and stored?

We thank the reviewer for bringing up this point. As we just mentioned, we have added information to that section of the introduction about spatial plasticity and flexibility that are needed to effectively map the environment. However, we are careful to not go into mechanisms/machinery in the introduction as our study does not address a physiological or neurological component. We do however go into more detail of proposed mechanisms and underlying machinery for spatial memory in the discussion.

I think the authors should move towards a better way of visualizing the data (Figs. 3 and 4) and show all the data. Currently the data are presented in a way that only allows readers to see means and SE, which many agree mask important individual variation (e.g., Weissgerber et al., 2015 PLoS Biology). I would encourage the authors to change the way the data are presented so readers can make assessments of the data themselves and visualize individual-variation.

We appreciate this point and are aware of the movement toward showing all data, and the debate that has been sparked by this movement. We agree that providing the raw data is important, especially when there is a risk of masked sub-populations within a data set. We strongly feel that this is why making the raw data available via data depositories or supplemental material (as we have done with this data set) is justified. However, it is also our experience that standard errors often capture the degree of variation fairly well, and that presenting means +/- SE is often an adequate way to visualize variation. This is why, as a matter of practice, all members of my lab visualize the raw (individual) data in addition to means/SEs, and we use this knowledge to inform the best way to finalize our figures for publication. When issues raised by Weissgerber et al. 2015 and the like are clear, we opt to present individual data as well. However, when those issues are not problematic for a data set, we consider the simplest, clearest, and most elegant way we can present the data. Indeed, one argument against presenting all individual data instead of or in addition to the means and SE (particularly for something like figure 3) is that the graph will be extremely unclear or uninterpretable. This defeats the purpose of presenting a figure in the first place. Again, we emphasize the raw data are fully available to the reader for anyone that might wonder if such masked effects exist that we did not consider.

For all of these reasons, and to address the reviewer's preference, we have changed figure 4 (which clearly do not mask 1 or more sub-populations), but it is our strong preference to maintain Fig. 3 as it is. We have also presented the individual data in the newly added figure (Fig 5) for the same reason as stated for Fig 4.

Other minor comments:

Line 85 – to provide readers with greater understanding of the motivation of this experiment, could you indicate how often OSR in prairie voles fluctuates in natural populations? Is it likely that male prairie voles even experience female- and male-biased OSR in nature? If they do experience a male-biased OSR, couldn't they just leave that area to find another?

Thank you for bringing up this important point. We have now added more context about how this experiment is appropriate given the natural variation of population and sex ratios in wild prairie voles, thus this being a very ecologically relevant cognitive demand.

Line 109 – are there any quantitative measures to show that the vegetation in the enclosures was the same?

All assessment was qualitative and unfortunately we do not have any quantitative measures of vegetation.

Line 124 – could you give a few more details on the trapping schedule used here? Randomly distributed traps?

Thank you for this point, which was also raised by reviewer 2. We have added more details about our trapping procedure.

Line 127 – here the location of those animals that were not recovered should be provided. In addition, the authors should indicate what the vole OSR and density was in each enclosure at the end of the 4 week experiment.

Done

Line 131 – are there data on how long it took to get the males from the lab to do this test?

We do not have data specifically measuring this duration, though typically within 2-3 hours of arriving in the lab from the field. We added this time estimate (2-3 hours) in the methods section.

Line 136 – is this thigmotaxis zone based upon some previous study? It seems arbitrary but this may just be my lack of familiarity here.

Thigmotaxis, or animals clinging to the wall, is important to measure because it denotes whether the subjects really 'understand' the task. If animals are spending the vast majority of their time along the walls this means the subject has yet to learn that no escape is offered along the perimeter. So this is a good index to see if the animal is actually problem solving via spatial navigation. (This is also covered in Vorhees & Williams 2006, now additionally cited after mention of thigmotaxis)

Line 153 – could you indicate 1) how many observers extracted behavioral data from the videos and 2) whether they were blind to the treatments?

Thank for pointing this out, we have clarified in the methods that we used an automated software package to collect these data. In other words, all observations were collected by computer software (EthoVision) and not by potentially biased human observers.

Line 161- I'm not quite understanding why sex is in this model. Were females tested?

This is an error due to shorthand. What was in the model was 'sex bias' of the enclosure and this has been corrected in the text.

Line 162 – these should all be t-statistics since the numerator df are 1?

changed

Line 166 – it would be helpful here to say something like “at the end of the trials, males from the MB found the platform x% faster than at the beginning whereas males in the FB enclosures ...” just to put some biological reality on the results.

Done

Line 174 – this generalized linear model needs to report the dispersion parameter and whether or not it was overdispersed.

We have added this reporting to the statistical analysis section in the methods.

Line 202 – but isn't it difficult to test either of these hypotheses without a third treatment where OSR was equal? Similarly, the lack of multiple replicates for the OSR treatments is a major confound.

Please see our responses to these questions above

Reviewer 4

This is an enjoyable paper that sets out to measure the effect of the social environment on spatial learning and memory performances. Specifically, the paper explores the effects of two different types of competition (contest versus scramble) on male spatial performances, by comparing two prairie vole groups, each exposed to a different sex-ratio treatment. The paper reports that males living in the male-biased sex ratio treatment outperformed males that have previously lived in the female-biased sex ratio treatment. The paper concludes that intensity of contest competition (male- male competition) causes superior spatial ability performances, whereas scramble competition (fewer females) does not.

I think this paper holds much promise and will be of interest to readers of RSOS.

Thank you for your positive impression of our paper.

However, I had some concerns regarding justification of the experimental design and the interpretation of the results. It is unclear that the female-biased sex ratio treatment represents scramble competition and the male-biased treatment does not? The definition given in the

introduction states that scramble competition “*involves males searching for females, particularly when the number of available females is limited.*” – Why are females *more* limited in the female-biased population compared to the male-biased?

In the introduction, we were giving merely one example in which scramble competition can be high, which is when there are very few females and thus tracking them effectively has huge pay offs for male reproductive success and huge costs if done poorly. However, we recognize now that highlighting this exemplar in the introduction is confusing given it is the opposite of what we designed. Instead we have now highlighted in the introduction a different ecological exemplar of scramble competition that is more congruent with the experiment we conducted and clarified what is involved in scramble competition.

We explain this in more detail below when specifically addressed in the discussion.

It is also not clear that the data support or allow you to distinguish between the two types of competition with your sex ratio treatments. Additionally, the interpretations of findings require toning down. Specifically, because the number of females was kept constant across the treatments and was not varied experimentally, the paper is not able to make inferences about the effect of searching for females on male spatial performances.

Thank you for raising these points, we have addressed them specifically below. Here in brief, we aimed to create scenarios where the intensity of each type of competition was varied. Thus, we have tailored the interpretations of our findings to more closely represent our manipulations.

The final concern is that the treatments differ in overall group size and the number of males – both of which, may be the causal factor in affecting spatial performances and this needs to be acknowledged.

We agree with this point, and we clearly failed to address this sufficiently; all four reviewers raised this point. We now explicitly justify why we did not account for population density and why we feel this potential confound was not a serious concern. But we also clearly acknowledge this issue in the discussion. Please see our more elaborate response to this criticism in our response to Reviewer 1 Point 1 above.

The specific statistical analyses conducted are unclear and the paper would benefit from a dedicated section about the analyses for each response variable and the tests used with justification for each. There is also confusion in the results section regarding which fixed effects were included in analyses. Specifically, it is unclear why sex was included as a fixed effect if only males were included in analyses?

We have modified the statistical analysis section of the methods to be clearer. We inappropriately used the term ‘Sex’ as short-hand for sex bias in treatment. We have corrected this in our explanation of the model.

Please see comments below regarding specific areas that need further clarity.

Abstract

It is unclear from the abstract how the different sex-ratio treatments relate to the two types of competition.

We have clarified this in the abstract.

Line 24: “Equally complex spatial context” – of the task or the environment?

We use this term to indicate that the experience of living in a natural outdoor field enclosure (regardless of OSR) is much more complex than living in a standard laboratory shoe box cage. We have clarified this point in the abstract.

Introduction I struggled a little bit with the opening paragraph, there are some nice examples and it does a good job of drawing the reader in, but it would benefit from a bit of rephrasing. Please see suggestions below:

Line 32: “*Navigating the world...*” use ‘environment’ instead of ‘world’, it is then consistent with the rest of the MS.

Done

Line 33: Change “...*spatial locations of contexts, presumably that lead to salient features of the world (like food sources or familiar territories)*” to “...*the location of salient features in the environment (e.g. food sources or territories)*”.

Done

“Lines 35-36: “*although some recent work has begun to consider these questions.*” – such as?

We have added a couple citations that hone in on the context of how space is actually used, and what that means for spatial cognition and social decision-making. For better or worse, this is as much as we can do to this end, largely because our lab group is currently at the forefront of bridging the literatures of spatial learning & memory with social contexts (specifically mating behavior). As a result, there is very little in the way of pre-existing literature on the topic of socio-spatial memory.

Line 44: use ‘Similarly,’ instead of ‘In another example,’

Done

Line 46: In what way is this an example of episodic-like memory? Is this ‘what, **when** and where’ important to your hypotheses or the story of the paper? If not, I wouldn’t use this term because it requires explanation, instead, just focus on Foley et al.’s findings that contribute to your questions/story.

Done. We have removed the term episodic-like memory to avoid confusion, and we agree it is not important to the hypotheses or story of the paper.

Line 54: References and some finessing of the definitions of the two types of competition would be helpful here. How does female availability relate to contest competition and is it important to the definition? Are there situations in which you can have contest competition and no scramble competition?

We thank the reviewer for this point. We have added more context in the section to better understand how scramble competition and contest competition relate to one another. Female

availability is important to the definition especially in the context of this experiment because we are trying to manipulate access to females by increasing male ratios and thus contest competition. This is another reason why we kept the number of overall females constant in both OSR contexts because the important factor is determining whether the prominent scenario of intraspecific competition created is contest or scramble is the number of males relative to females.

As we note in the Introduction, male reproductive success has elements of both tracking the movements of females and tracking the movements of competitors. Mating presents an interesting challenge for males where they encounter both types of intraspecific competition to be successful. Therefore it is outside of the scope of our study to conclude whether one type can occur completely separate from the other. The goal of our study was to determine which type of competition more strongly drives spatial cognition in males, given that males inevitably need to effectively manage both.

Line 58: resources that attract who? The males or the females? Please rephrase.

Done

Line 60: This line states that intensity of intraspecific competition (male-male competition) influences male-male competition. Please rephrase.

Done

Lines 70-72: *“Taken together, variation in OSR can establish social contexts in which variation in behavioural tactics (such as how to approach intraspecific competition) reflect solutions to those immediate contextual challenges”* – it is unclear what it meant by *“how to approach intraspecific competition”*, what behavioural tactics exactly are you referring to?

Here we were referring to the payoff matrix of costs and benefits associated with intraspecific competition mentioned prior to this concluding sentence. We agree that the parenthetical statement was quite vague and unclear. Instead we have changed it to “(such as whether to prioritize tracking competitors vs. potential mates)” to eliminate any further confusion.

It would be nice to know a bit more about the natural history of the voles. How are females distributed in the environment? How does a female choose a territory and then how does she choose a male? What are the average male and female home range size? Is there much reproductive skew in natural populations? What are the natural levels of competition between males and what does this consist of? Physical contests? Scent marking? Female mate-guarding?

We thank the reviewer for their appreciation of natural history. We added a little more context about how this experiment is appropriate given the natural variation of population and sex ratios in wild prairie voles. We also have included more context and references on general male and female mating behavior and competition.

Although we do know a considerable amount about prairie vole ecology and natural behavior, there is much that has not been studied or determined, including many of the questions you raised. For example, we do not know how males or females choose a territory (and if one or both sexes establish the territory). We also do not know how females choose males. In fact, very little has been written on prairie vole mate choice, and our studies (published and

unpublished) have found that females do not seem to have stereotypical characteristics on which females base their preferences (see Ophir et al. 2008 BBE, for the only published study of female prairie vole mate choice of which we are aware). We also note that some of this information is only loosely relevant to the current study. For example, we did not measure home range sizes, however, territory sizes vary widely based on the population size and individual mating tactic. We are unaware of studies that have measured natural reproductive skew in prairie voles, or of studies that have tested variation in the nature of competition between males (outside of this manuscript). As a monogamous species, these animals have not been the focus of classic sexual selection studies that characterize male-male aggression, though we do know they engage in aggression against intruders in the lab. Scent marking has been studied (more in the congener the Meadow vole, but also in prairie voles), but we do not see the relevance of this work to our study. Female mate guarding has been looked at indirectly through patterns of space use, and inferred from resident intruder lab tests, but again this has not been studied directly. The bottom line, is that the reviewer raises many very interesting questions, and highlights many unanswered and important questions that remain.

Methods Line 91: “...unrelated pairs of F1 **or** wild-caught...” – how many did you have of each? Or do you mean that F1’s are your wild-caught individuals?

We have further clarified this statement; Reviewer 1 also expressed confusion to this sentence. This statement was referring to our breeding colony in the lab. We had a colony of 15 breeders at the time consisting of some wild-caught and some wild-caught/F1 pairs. All 15 breeding pairs were unrelated, and the subjects used for this experiment were offspring from those breeders. The new phrasing in the methods aims to make these points clearer.

Line 92: change to ‘old’

Done

Line 94: Please specify what dimensions these measurements represent

Done

Line 97: Cool. I know nothing about ear-tagging voles! What materials do you use? Are they numbered/coloured/symbols? Please elaborate.

The ear tags are small, metal, and laser etched with a unique four digit ID. All of this information has been added to the methods.

Line 109: delete “...directly comparable and...”

Done

Lines 113 – 117: I appreciate that the male-female ratio’s were kept the same, but reversed, across the treatments. However, I am unclear why it was more important to do this than standardising overall group size? There is now an added confound, as group size (with no mating competition) has been shown to have a positive causal effect on cognitive performances in an avian species (Langley et al. (2018). *Animal Behaviour*, 142, 87–93). Similar processes could also affect the voles. It therefore makes it difficult to conclude that it was a difference between the treatments in male- male competition that caused the differences in spatial ability. It could simply be caused by a difference in the overall number of individuals.

We obviously overestimated how well we dealt with the issue of population density in our original draft and thank the reviewer for helping us realize this. Indeed Reviewers 1, 2 and 3 also noted this issue. Rather than recap our response at the risk of being redundant, we refer the reviewer to our response to Reviewer 1 (Point 1), where we tried to clarify this issue better. Briefly, we did consider this when we were designing the experiment, and chose to design our experiment this way based on both theoretical and empirical support that population density is highly unlikely to influence the results, whereas altering the number of females per unity space would pose a more serious confound to our design. We admit that we should have been much better about explaining the justification for this design and providing a discussion of this at the end of the paper. We have now done both on in the Methods and Discussion.

Line 118-121: This is a bit unclear. Are you arguing that female distribution is relatively uniform across the two treatments and therefore the only difference is male-male competition and the number of females? It could still be a function of density – I'm not sure the data allows you to rule that out as a possible explanation.

Yes, that is precisely what we are arguing. Given a relatively uniform distribution of females across treatment, the main manipulation is number of males relative to a standard number of females, and the intensity of intraspecific competition results from the altered male-male competition. As stated above, we agree that we cannot rule out density and have included this explanation further in the methods and discussion.

Line 126-127: please give the survivors for each sex within each treatment.

Done

The American spelling of 'behavioural' is used throughout the MS. I'm unsure what the journal prefers?

We thank the reviewer for pointing this out. We are also unsure which spelling is preferred (and if this is for all language or just certain words as it the case for some journals – like Animal Behaviour). Because we are more accustomed to 'American' English spelling, we (embarrassingly) rely on Copy Editors of the journal to help with such issues.

Line 131 – were subjects tested individually? Please be clear.

Yes, the Morris water maze requires each animal to be tested individually. We have made this point more clear in the revised manuscript.

Lines 133-141: It would be helpful to describe the apparatus first and what the animals were required to do and then describe your zones and their measurements.

Done

Line 146: Were all 10 learning trials done on the capture day? What was the inter-trial interval? This is particularly important for the memory trial – how long after trial 10 was the memory trial conducted? Ensure methods are reproducible for the reader.

Thank for bringing these points up as it allows us to add more clarity to the methods section.

The 10 learning trials occurred over 5 days, with 2 trials per day. For each day the inter-trial period was 1 hr. There was a 1 hr. inter-trial period between the 10th trial and the memory trial. This has all been added to the revised manuscript.

Figure 2: I found the arrows on the diagram a bit confusing; it looks as if when you get to thigmotaxis zone near the platform, they overlap, but I realise these arrows are not to scale. Perhaps using dashed lines to draw the zones (approximately) over the photo would be helpful.

We apologize for any confusion; however, the zones are already drawn on the image (with solid lines). The arrows are simply labels for these zones. The thigmotaxis zone (which is highlighted in a lighter shade to make it obvious) and 'near platform zone' do not overlap, and this should be clear from the figure and the boundary lines drawn to demark each zone. We have modified the figure caption to clarify this better.

Figure 4: Is this the spatial **memory** task? Please include in figure legend.

Yes, this is the memory task. We have added memory trial to the figure legend.

Results

Throughout the results 2dp and 3dp have been used. There are also some formatting differences, such as a space between df in some cases but not others; sometimes there are commas between each test statistic and sometimes not. Please be consistent.

Thank you for raising this concern, we now have made our formatting of results consistent throughout.

The paper requires elaboration on the statistical analyses used and I would suggest including this as a separate section at the end of your methods section and to walk the reader through which analysis was conducted for each question. That way, discussion of methods can be removed from the results so that **only** results are stated here. In the methods section it would be useful to include the following: What software was used?

Was this a Gaussian distributed mixed model? Were your model residuals normally distributed? If not, you may need to think about the appropriate error structure to use as latency data is characteristically poisson distributed. Please justify your choice of model for **all** questions. How were your F statistics generated? Are they from comparisons of model deviances? Any model simplification? Please explain.

Thank you for this suggestion. Yes the model was a Gaussian distributed mixed model. The residuals were normally distributed. The errors were also normally distributed and we had homoscedasticity. The F statistics were generated using Type III sum of squares. The comparisons were from model deviances and no model simplifications were used.

We have now added a statistical analysis section and removed all discussion of the model from the results and placed it into this section of the methods. We now justify more specifically why the models we ran were appropriate for the questions we were asking. We also report details regarding the distributions and dispersions of our models.

Line 159: This is the first mention that females were tested – your hypotheses are purely based on male cognition in relation to differences in male-male competition/locating females. So it is

unclear why 'sex' is included as a fixed effect? The paper has not explained why females are included in analyses and what differences we'd expect to see.

Thank you for raising this, another reviewer as caught this error. This was shorthand for "sex bias" and has been corrected in the revised manuscript to eliminate confusion.

Line 161: How have you "...controlled for..." swim velocity and thigmotaxis? Please explain, were they included as a fixed effects?

Yes, we included both swim velocity and thigmotaxis as fixed effects, this has now been explicitly mentioned in the methods section describing the statistical models.

Line 164: In the description of the analyses above there is only mention of a sex * trial interaction...not treatment * trial. Please ensure the results match what is described above.

Thank you for catching this inconsistency. We have change 'trial x treatment' to 'trial x sex bias' to remain consistent with the prior description of model.

Line 177: first mention of proximity to the goal being narrowed. Please explain.

In our earlier explanation of the memory trial in the methods we explicitly say, "we compared the number of visits to, and duration of time spent swimming in, the vicinity to the platform's original location". We agree that the wording of "narrowing the proximity" was confusing. We simply used the phrasing as a transition from discussing the results in the platform quadrant to the smaller zones measured. We changed the sentence to, "When narrowing our focus on the subjects' swimming closer in proximity to the goal" to make this intention more clear.

Lines 179-181: This belongs in methods

Moved to methods

Discussion

Lines 195-196: "...males must rely on their spatial memory to locate potential mates, particularly when females outnumber males and conditions are ripe for intense scramble competition..." – the definition in the introduction states that scramble competition occurs when females are limited. In this case, females are not limited. Please explain.

Thank you for pointing out this discrepancy. The section in the discussion is correct, and we restated the point in the introduction to be clearer. The aspect of scramble competition we are mimicing in the experiment is the need for males to track potential mates in space (rather than track competitors). Thus, having females outnumbering males in the female biased OSR sets up that paradigm.

In the introduction, we were giving merely one example in which scramble competition can be high, which is when there are very few females and thus tracking them effectively has huge pay offs for male reproductive success and huge costs if done poorly. However, we recognize now that highlighting this explemplar in the introduction is confusing given it is the opposite of what we designed. Instead we have now highlighted in the introduction a different 'real-world' example of scramble competition that is more congruent with the experiment we conducted.

The number of females is identical across the treatments (scramble competition has been 'standardised'), yet contest competition has been varied. Therefore, the data justify the discussion that contest competition may be driving differences in spatial performances but the data does not allow you to comment on the effect of scramble competition because this has not been varied. Some clarity is needed throughout the discussion and it feels like there is new justification of hypotheses in the opening of the discussion that really should have been detailed in the introduction and then summarised here.

Standardizing the number of females does not standardize the potential for scramble competition. The relative number of females to males will impact which form (or intensity therein) of competition will arise.

Furthermore (and to this point), we are very clear in the discussion that our hypotheses about how male mating behaviors express themselves under these two conditions (the form of competition) are not necessarily mutually exclusive expectations. Our intent was never to completely eliminate one type of competition (it is probably not possible). Rather, we aimed to change the relative intensity of each type of competition. As the Reviewer stated, we did indeed standardize female numbers (in hope of standardizing female distribution), and we altered the number of males relative to the number of females. This should have had the desired effect of increasing the intensity of scramble competition in the FB enclosure, because there were fewer male competitors for each available female. Likewise, the intensity of contest competition should have been higher in the MB enclosure because there were more male competitors relative the number of females in that context.

When we contrast the contexts of contest vs. scramble in the discussion (and introduction), we are very careful to emphasize that we were attempting to manipulate the relative pressure on males to either track other competitors more or potential mates more (based on the relative number of males to females). We feel that our data do indeed speak to this question, and justify the claim that the pressures associated with (relatively) high intensity contest competition are more influential in shaping spatial performance than a context with (relatively) more intense scramble competition.

It would be nice to open the discussion with a summary of your findings, then go on to explain how these findings support or refute your hypotheses (which are detailed in your intro) and fit into existing literature.

Good suggestion. Change made.

Line 197: It is correctly stated that the two potential mechanisms for a difference in spatial ability are non-mutually exclusive. It is unclear how the findings allow you to separate them. Please explain.

We have explained this above in our response about standardized scramble competition.

Lines 200-202: "Specifically, we found that males exposed to a male-biased environment in which competition for living in a social context in which males outnumbered females outperformed males living in a social context in which females outnumbered males." – this sentence is a lot to digest, a few commas may help to break it up and it'll be easier to process. It would also be good to include how they outperformed the other males, i.e. spatial learning and

memory tasks.

Done. Added commas and clarified that they out-performed on spatial learning and memory tasks.

Line 205: “*This result is consistent with other work that found male prairie voles attend to the social identity of males more closely than females (Zheng et al. 2013)*” – I’m not sure on the relevance of the comparison here, in what way do males attend more to males than they do to females? Please explain.

This is relevant because our whole paradigm is whether males should attend to (i.e. track) potential mates vs. competitors. Zheng et al. 2013 is a lab study that also demonstrates that males are paying more attention to their potential competitors than potential mating opportunities (i.e., demonstrate social recognition for males but not females). It is a matter of whether males or females have more salience and relevance to the individuals under study.

Line 214: Please include “*spatial learning and memory*” throughout the manuscript (unless of course discussing the individual measures) or refer to them collectively as “*spatial ability*”

Done

Line 217: Cognitive or behavioural ‘flexibility’ (more commonly measured using reversal learning or inhibitory control tasks) was not assessed in the paper, so I would be careful with the wording here.

We have been more careful in our wording. We simply meant flexibility in that differences in spatial ability between the groups, arising from the field manipulation, occurred after only 4 weeks of treatment suggesting that the mechanisms (which you are right we did not measure) of spatial ability are plastic and flexible. We also note that in a departure from the traditional usage, the term flexibility is increasingly being used to refer to variation within an individual. Nevertheless, we take the Reviewer’s point.

Line 219: “*The form that such a response should take...*” – What response is this? Response to social composition/sexual competition or their cognitive performance? Please explain.

This is in the sentence immediately preceding it “spatial flexibility in performance might enable animals to respond”. Thus the response referred to in the very next sentence is one of spatial ability/cognitive performance. We have made this clearer in the revised version.

Line 222: delete “*...(and perhaps should)...*”

Done

Line 231: just spatial memory or spatial learning too? The opening sentence discusses memory specifically but the examples given regard both learning and memory. Please be clear.

Done, changed to “learning and memory”

Line 242: “*...the performance of all our subjects should have been equally strong.*” – as strong as what? Each other? Do you mean that there would have been no difference in spatial learning and memory between the treatments because all other factors were identical and only the social

composition differed?

Here we meant equally strong compared to each other. Yes, this is exactly what we mean. We have added “compared to each other” after equally strong.

Line 245: delete “...(not spatial)...”

We have deleted (not spatial), however we replaced it with “rather than the”. We feel very strongly that it is important for us to drive this message home that it was the altering of social complexity, in the absence of altering spatial complexity, that led to our observed results. Therefore we prefer to keep this distinction in the text.

Line 248: In what way are these environments ‘spatially complex’? – There is not much description in the methods of this complexity. It is not overly important to state this anyway, as this is not something that was varied experimentally, it is simply important to highlight that the treatments were spatially identical and varied only in their social composition.

In a few sentences before this statement we detailed the complexity of living in nature, “all of the animals in our experiment were equally exposed to the challenges of living in nature, including experiencing temperature fluctuations and a need to locate shelter, the need to explore their environments and forage for food, and experiencing diverse terrain with a heterogeneous mix of vegetation and spatial landmarks”. Thus, it was implied in the following sentences that our subjects faced spatial complexity compared to being in a standard lab cage. We have made this more explicit with the following addition. “In comparison to laboratory housing, these challenges were much more complex, and in particular profoundly more spatially complex.”

Lines 249-250: “To our knowledge, this is the first demonstration that social context, while holding spatial complexity constant, impacts cognitive functions related to spatial information.” – this is not entirely true, social context (in terms of group size and individual social interactions) has been shown to influence spatial learning in other taxa. Please see:

- Langley et al. (2018). Individuals in larger groups are more successful on spatial discrimination tasks. *Animal Behaviour*, 142, 87–93.

- Barnard, C. J., & Luo, N. (2002). Acquisition of dominance status affects maze learning in mice. *Behavioural Processes*, 60, 53–59. doi: 10.1016/S0376-6357(02)00121-3

Thank you for pointing these out. We think these papers are very interesting, but did not quite demonstrate what we have found. Firstly, our study was conducted in the field (which is important), and we have now added that distinction to this sentence. Secondly, we think that these studies did not quite manipulate social context because neither group size nor individual experience establishing dominance is the same thing as creating different social contexts, as we have done. However, to be clearer, we make the point to highlight that our results are focused on group composition (i.e., relative number of males to females), which is clearly not what these other papers manipulated.

Line 252: The spatial environment was not manipulated so I would remove “spatially complex” here.

We have clarified this in the modified final sentence and think our point is clear that we did not manipulate spatial complexity. However, we purposefully emphasize the effect was attributable

to social context (and not spatial complexity), because there is a lot of data that would lead some readers to interpret our results as a function of manipulating spatial complexity (lab to field changes). This is why we included the section on this topic in our discussion. We feel it is important to highlight this contrast for fear that readers will misattribute our results to spatial complexity of the field environment.

Appendix C

Reviewer comments to Author:

Responses to Reviewers in Blue

Reviewer: 1

Comments to the Author(s)

The authors have done a very good job at revising their manuscript in response to the substantive and lengthy suggestions made by reviewers. I applaud their dedication to detailing their revisions.

I feel that the manuscript is much improved, and the authors have satisfactorily addressed most of my comments. Specifically, the new additions to the text in the methods and discussion regarding the experimental design help the reader understand the potential limitations and complexity of the study. I think that the new sections in the discussion are particularly helpful additions.

However, I still have one issue that the authors should consider more fully. (All line references below refer to the tracked changes version.) I remain concerned that the FB group did not demonstrate learning in the water maze. I appreciate that the authors are clear with their goals in their response – they did not intend to test if the voles could reach a specific criterion or threshold. This is fine. However, they never demonstrated that the FB voles learned the maze and so their subsequent test (trial 11) of that group is irrelevant.

What I find most confusing here is the authors' claim that the FB group did learn the maze because the slope of the line in figure 3 is not zero and that there was a difference in performance between trials 1 and 10. I do not agree with this interpretation at all. Both of these criteria could potentially hold as a result of random variance if the voles indeed did not learn. The statistical analyses of those differences do not suggest learning. In fact, on line 262, the author state that the "FB males showed no difference in latencies across the learning trials" and provide statistics to support that point. Then, on line 264, the authors state that the FB group found the platform 17.5% faster, but this includes no statistical analysis. An anecdotal view of figure 3 suggests a great deal of overlap of the SEM, and so I doubt that this comparison is statistically different. So, in any other situation, one would interpret that the FB voles did not learn.

Although these voles are not in a situation of "profound memory or learning impairment" and the assumption is that they should be able to learn, this result still raises a red flag for me. On one hand, it is possible that the authors found a test with a complexity and difficulty level that exactly splits the two groups at their threshold (and thus resulting in one group that could learn and the other that could not). However, because the authors did not demonstrate that the FB group could learn, i.e., no positive control, they cannot dissociate whether the FB group's learning was not so good as the other group, if the Morris water maze was an inappropriate task for this group, or if something else might have been at play preventing them from performing in the maze that is independent of

spatial memory abilities all together. Without this verification/control, one could speculate all sorts of things. Perhaps the males in that group were fatter (with more access to food resources with fewer individuals in the enclosure) and so could not swim as well.

My point here is not to suggest that the authors propose or address alternatives. I admit that the probability that my concern is creating a spurious effect is relatively small. Indeed, all of the arguments that the authors provide are valid arguments and the experiment itself helps reduce some of those concerns. However, from the perspective of experimental design, interpretation, and transparency, the authors really should address this caveat and admit that they cannot definitely rule out the possibility, however unlikely.

We thank the reviewer for comments regarding this area of our manuscript. We can see as you pointed out, how some of our explanation of the results appears contradictory. To rectify this, and because we agree that we cannot entirely rule out the FB males' inability to statistically learn the task, we have modified our language in the results section.

Specifically, we now state: "Although we cannot rule out the possibility that males in the FB context demonstrated learning impairment, we note that by the end of the learning trials (trial 10), FB males found the platform 17.5% faster than the initial trial (trial 1), whereas MB males found it 65.3% faster. Thus, males in both contexts demonstrated a consistent ability to locate the platform with some degree of improvement. However, notably, MB males clearly showed significant improvement in spatial performance over the course of learning trials, whereas FB males did not demonstrate learning to the same degree."

We believe these changes address this important issue for which we no longer attribute statistically valid learning to the FB group, and instead make the simple observation that the FB males can find the platform every trial and demonstrate the modest qualitative improvement in FB males, and stronger improvement in MB males (which is the effect upon which we wish to focus the readers attention).

Additionally, in the discussion section we have toned down our interpretation by making the correct noted below.

Line 389. I suggest that the authors are a bit more tentative here with the language and replace "indicate" with "are consistent with the hypothesis that".

DONE

Reviewer 3
Comments to the Author(s)

The authors have made numerous revisions to this manuscript after 4 thorough reviews. I'm very satisfied with the changes they have made and their detailed and thoughtful responses. My only comment here is that in some places (e.g., line 224) they are providing the numerators degrees of freedom for t statistics (=1) whereas others they are not (e.g., line 235). I think you do not need to indicate numerator degrees of freedom as 1 in these places such as line 224.

We thank the reviewer for their comments, especially the recognition of all the revisions we have made. We believe that this manuscript has been greatly improved by incorporating the feedback from the reviewers.

We have removed the numerators of degrees of freedom in the reporting of all of our t statistics in the results section of the manuscript. Thank you for catching this mistake.